# Japanese encephalitis virus activates the NLRP3/caspase-1/GSDMD signaling pathway in dopaminergic neurons

Xiaoyan Guo[iD], Siyuan Lu, Zhiwei Huang, Jiahuan Li, Guofu Cheng, Xueying Hu, Wanpo Zhang, Changqin Gu[iD]*

College of Veterinary Medicine, Huazhong Agricultural University, Wuhan, Hubei, China

* guchangqin@mail.hzau.edu.cn

## Abstract

Japanese encephalitis virus (JEV) preferentially targets brain regions rich in dopaminergic neurons, including the thalamus, midbrain, and striatum, leading to severe neuroinflammation. However, the underlying pathological mechanisms remain unclear. This study hypothesizes that JEV-induced pyroptosis exacerbates neuroinflammation by activating the NLRP3/caspase-1/GSDMD signaling pathway. In vivo experiments using JEV-infected mice revealed virus-specific targeting of dopaminergic neurons, concurrent activation of the NLRP3 inflammasome, and induction of inflammatory responses. In vitro studies with mouse midbrain dopaminergic cells showed significant viral replication, progressive cell membrane damage, and colocalization of JEV with pyroptotic markers. The pharmacological inhibition of NLRP3, while suppressing viral load and NLRP3 inflammasome activation, fails to delay the onset of neuroinflammation, potentially due to the activation of glial cells. This suggests that glial cell-mediated inflammatory pathways independent of NLRP3 may drive neuroinflammation despite NLRP3 inhibition. The unimpeded activation of glia could sustain pro-inflammatory responses, highlighting the need to target glial activation alongside NLRP3 to effectively mitigate neuroinflammation.

## Author summary

Japanese encephalitis virus infection can induce severe neurological symptoms, including tremors, slow movements, and motor impairments, indicating severe infection of dopaminergic neurons. These symptoms primarily occur in the later stages of infection, suggesting that sustained inflammatory responses may be the primary cause of neuronal damage. Our research team has previously confirmed significant upregulation of genes related to NLRP3, caspase-1, GSDMD, and other molecules after infection, indicating their involvement in neuroinflammation. Pyroptosis, as a mechanism of inflammatory cell death, has been shown

**Data availability statement:** All relevant data are within the manuscript and its Supporting Information files.

**Funding:** This work was supported by grants from the National Key Research and Development Program of China (SQ2022YFD1800105) and the National Natural Sciences Foundation of China (32030107) to CG and SC. The funders had no role in the study design, data collection and analysis, decision to publish, or preparation of the manuscript.

**Competing interests:** The authors have declared that no competing interests exist.

to participate in many viral infections, but there is no specific evidence supporting its exact pathogenic mechanism in JEV-infected neurons. To determine the role of pyroptosis in JEV infection, we conducted cell infection and animal infection experiments. After confirming the crucial role of NLRP3, corresponding inhibitors were used in regression experiments, confirming that inhibiting the NLRP3-related signaling pathway effectively alleviates the clinical symptoms induced by the virus.

## Introduction

Japanese encephalitis is a zoonotic disease transmitted by mosquitoes harboring Japanese encephalitis virus (JEV) [1]. Studies indicate that over 2 billion people reside in endemic regions worldwide, with approximately 50,000–175,000 reported cases of JEV infection annually. The case fatality rate ranges from 20% to 30%, whereas neurological deficits affect 30% to 50% of infected individuals [2], suggesting that JEV directly targets neurons, leading to mortality. Notably, no specific antiviral therapies exist for Japanese encephalitis, necessitating reliance on supportive care to manage symptoms and complications. Vaccination remains the cornerstone of prevention, with demonstrated efficacy in reducing disease incidence. JEV involves a complex interplay of pathological mechanisms characterized by endothelial activation, microvascular injury, and immune cell infiltration. Key cellular targets include endothelial cells, microglia, and neuronal progenitor cells, with neurons serving as primary hosts for viral replication due to their tropism [3,4]. Infected neurons initiate apoptotic pathways that propagate neuroinflammation [5], while microglial activation triggered by low viral loads accelerates the release of pro-inflammatory mediators such as cytokines and chemokines [6]. This creates a self-perpetuating inflammatory loop within the neurovascular unit, contributing to persistent blood-brain barrier dysfunction and immune cell recruitment.

Viral encephalitis primarily targets the central nervous system (CNS), exhibiting distinct tropism and persistence patterns across different brain regions. This anatomical specificity underlies the heterogeneous clinical manifestations, ranging from mild neurological deficits to severe life-threatening complications. Moreover, disruptions in neurotransmitter homeostasis within the affected neural circuits play a pivotal role in precipitating a spectrum of neurological dysfunctions, including altered states of consciousness, cognitive impairment, and motor disabilities [7]. Dopamine, a crucial neurotransmitter secreted by dopaminergic neurons (DA neurons), extensively projects through the mesencephalic limbic and cortical dopamine systems, predominantly concentrated in the midbrain [8]. When the substantia nigra and striatum are damaged, motor dysfunction, including tremors, rigidity, and bradykinesia, resembling the symptoms of back bowing, tremors, and hind limb paralysis observed following JEV infection [9]. A study on the neuronal invasiveness of JEV demonstrates that JEV selectively targets specific neuron types [10]. including dopaminergic neurons, which are prevalent in the thalamus, midbrain, and striatum. This finding indicates a strong

affinity of JEV for dopaminergic neurons [11]. JEV infection may lead to motor dysfunction, which is characterized by the impairment of dopaminergic neurons.

Recent evidence highlights pyroptosis as a critical cell death pathway in JEV-infected macrophages [12,13]. This inflammatory form of programmed cell death is distinguished by Cysteinyl aspartate specific proteinase (Caspase-1) activation, Gasdermin D (GSDMD) pore formation, and release of Interleukin-1β (IL-1β) and Interleukin-18 (IL-18). Immunofluorescence studies demonstrate significant co-localization of GSDMD-N and TUNEL in infected macrophages, indicating dual apoptotic-pyroptotic cell death mechanisms. The upregulation of canonical pyroptosis-related genes further validates this novel pathological mechanism. Notably, while the involvement of immune cells in JEV-induced inflammation is well-documented, the mechanisms governing direct neuronal inflammatory responses remain poorly understood. This knowledge gap is critical for elucidating disease pathogenesis, as neurons are the primary targets of JEV. Neurons may activate intrinsic inflammatory pathways, such as the Nucleotide oligomerization domain-like receptor family pyrin domain containing 3 (NLRP3) inflammasome-pyroptosis axis, independently of immune cell-mediated cascades. These pathways could potentially serve as the primary drivers of viral neurotropism and subsequent neurological damage, highlighting the urgent need for further investigation into neuron-specific inflammatory responses during JEV infection.

Despite prior research linking Japanese encephalitis virus (JEV) to pyroptosis-mediated inflammation, whether JEV directly induces pyroptosis in neurons remains a critical, uncharted scientific question. Answering this query is pivotal for advancing our understanding of viral neuroinvasion mechanisms, as existing studies predominantly focus on immune cells' roles during JEV infection. However, neurons–primary targets of JEV–may independently activate the pyroptosis pathway, yet this potential mechanism remains largely unexplored. Our study aims to fill this knowledge gap by systematically investigating neuronal pyroptosis in JEV-infected dopaminergic neurons using *in vitro* and *in vivo* models, with a focus on that determining the occurrence of pyroptosis in these specific neurons, clarifying the underlying molecular mechanisms, and evaluating the functional consequences on neuroinflammation and neuronal survival. Uncovering the pyroptotic pathway in JEV-infected dopaminergic neurons will not only reveal novel virus-host interaction mechanisms but also identify potential therapeutic targets for JEV-associated neuroinflammatory disorders, thereby reshaping the understanding of JEV pathogenesis and paving the way for innovative neuron-protective therapies.

## Methods and materials

### Ethical statement

This study was conducted within the Guidelines of Regulations for the Administration of Laboratory Animals (Reversion of the State Science and Technology Commission of the People's Republic of China on March 1, 2017). All animals used in this study received prior approval from the Hubei Provincial Experimental Animal Manage Committee and Huazhong Agricultural University Academic Committee. The Hubei Provincial Laboratory Animal Quality Certificate number was HZAUMO-2022–0100, which was approved by the Scientific Ethics Committee of Huazhong Agricultural University.

### Cells and virus

Mouse microglia (BV2) were donated by Prof. Shengbo Cao of the State Key Laboratory of Agricultural Microbiology, Huazhong Agricultural University. The mouse midbrain dopaminergic neuron cell line (MN9D) was purchased from Cell Health, Wuhan, China. BV2 and MN9D cell were cultured in Dulbecco's modified Eagle's medium (DMEM; Gibco) supplemented with 10% fetal bovine serum (FBS; Gibco) at 37 °C in a 5% $CO_2$ humidified atmosphere. The P3 strain of JEV was donated by Prof. Shengbo Cao of the State Key Laboratory of Agricultural Microbiology, Huazhong Agricultural University.

### Mouse infection

Thirty BALB/c female mice aged 6 weeks were purchased from the Experimental Animal Center of Huazhong Agricultural University and raised at the Experimental Animal Center of Huazhong Agricultural University. The mice were divided

into a control group (n = 15) and a JEV infection group (n = 15). Each mouse in the infected group was intraperitoneally injected with 0.1 mL ($10^5$ PFU/mL), while those in the control group were intraperitoneally administered 0.1 mL of DMEM. Five mice in each group were dislocated and dissected at 3, 6 and 9 dpi, and the right brain tissue was removed and fixed in 4% paraformaldehyde for paraffin section preparation. The remaining brain tissue was frozen for subsequent experiments. Subsequently, A total of forty BALB/C mice were subsequently generated on the basis of our previous studies which included a control group (n = 5), the NLRP3 inhibitor group (n = 5), JEV infection group (n = 15), and JEV with NLRP3 inhibitor group (n = 15). Starting from 3 days before infection, mice were intraperitoneally injected with the NLRP3 inhibitor MCC950 (S7809, Selleck.cn, China) reconstituted in sterile PBS at a dose of 50 mg/kg per day. On the day of infection, mice were first injected with MCC950, and the virus was administered 4 hours later. The injection was continued for 9 days after JEV infection until tissue harvesting, death, or recovery of the mice. To assess the Neurological Motor Severity Score (NMSS) in mice from JEV, DMEM, and MCC950 treatment groups, the trial followed the same protocol as previous studies. Mice were continuously monitored for NMSS from 1 to 9 dpi, with the trial concluding on day 10 post-infection when 48 female BALB/c mice were euthanized. Paralysis severity was graded on a 0–5 scale: 0 indicated normal ambulation, and 5 denoted severe weakness in both forelimbs and hindlimbs. This grading system enabled quantitative assessment of paralysis in each mouse, with scores incremented by 0.5 for moderate symptoms.

## Preparation of paraffin sections and hematoxylin and eosin staining

The fixed brain tissue was removed and put into an automatic tissue dehydrator (TP1020, Leica, Germany) for gradient alcohol dehydration, xylene transparency, and wax dipping, and the section surface was embedded in paraffin wax to make tissue wax blocks. Serial sections were started along the median sagittal plane of the brain at a thickness of 4 μm for hematoxylin and eosin (HE) staining, immunohistochemical staining and indirect immunofluorescence staining. HE staining was performed with a fully automatic dyeing and sealing machine (ST5010 Autostainer XL and CV5030, Leica, Germany).

## Antibodies

Mouse anti-JEV-E was provided by the State Key Laboratory of Agricultural Microorganisms; mouse anti-tyrosine hydroxylase (GB12181, Servicebio, China); rabbit anti-IBA-1 (10904–1-AP, Proteintech, China), rabbit anti-GSDMD (#39754, CST, America) and mouse anti-caspase-1 (sc-56036, Santa Cruz, America), rabbit anti-Glyceraldehyde-3-phosphate dehydrogenase (GAPDH) (10494–1-AP, Proteintech, China); rabbit anti-caspase-1 (PAB46174, Bioswamp, China) and mouse anti-caspase-1 (sc-56036, Santa Cruz, America), rabbit anti-tyrosine hydroxylase (PAB37976, Bioswamp, China); rabbit anti-NLRP3 (WL02635, Wanleibio, China) and mouse anti-NLRP3 (sc-134306, Santa Cruz, America), rabbit anti-Apoptosis-associated speck-like protein containing CARD (ASC) (WL02462, Wanleibio, China) and mouse anti-ASC (sc-514414, Santa Cruz, America); rabbit anti-IL-1β(A9440, ABclonal, China), anti-rabbit IgG (H + L)-FITC (AS001, ABclonal, China) and sheep anti-mouse IgG (H + L)-Cy3 (AS007, ABclonal, China); goat anti-mouse IgG (H + L) (S0B4006, STARTER Antibodies, China) and HRP-conjugated goat anti-rabbit IgG heavy chain (AS063, ABclonal, China).

## Immunohistochemical staining

The slices were dewaxed, hydrated and rinsed under running water for 5 min. The sections were placed in citrate buffer subjected to microwave thermal repair, maintained above 95°C for 20 min, and then cooled naturally to room temperature. Then, peroxidase was used for blocking, goat serum was used for blocking, and primary antibodies (NLRP3 1:200; ASC 1:200; JEV-E 1:100; IBA-1 1:500; GSDMD 1:200; caspase-1 1:200) were added overnight at 4°C. After the samples were washed and incubated with HRP-labeled goat anti-mouse/rabbit IgG, DAB color development, hematoxylin restaining, hydrochloric acid fractionation, ammonia blue return, gradient alcohol dehydration, xylene transparency and neutral resin blocking were performed.

## Double immunofluorescence labeling

The sections were incubated with primary antibodies from two different sources, mixed proportionally (JEV-E 1:100, IBA-1 1:500, TH 1:200, GSDMD 1:200 and caspase-1 1:200) and incubated overnight at 4°C. On the next day, the primary antibodies were removed, and the samples were washed three times with PBS, 5 minutes per wash. Subsequently, the samples were incubated with fluorescent secondary antibodies for approximately 30 minutes. After discarding the secondary antibodies, the samples were washed three times with PBS (5 minutes per wash), followed by incubation with ready-to-use 4',6-diamidino-2-phenylindole (DAPI) staining solution at room temperature for 10 minutes. The smear was incubated with DAPI ready-to-use staining solution for 10 min at room temperature. After the anti-fluorescence quenching mounting agent was added, the staining was observed via a microscope (DM2500, Leica, Germany). Five randomly selected microscopic fields were captured via a 400 × light microscope. Images were collected via a confocal microscope, and the numbers of double-labeled cells were counted.

## Cell culture and infection

BV2 and MN9D cells were cultured in 12-well plates at $2 \times 10^5$ cells per well. Two groups (the JEV infection group and the DMEM control group) were set up with three wells in each group. When the cells reached 80% confluence, they were washed with DMEM 3 times and inoculated with JEV at an MOI of 5, after which the supernatant containing the virus was discarded after 1.5h of incubation at 37°C. Then, the cells were washed with DMEM 3 times, and 1 mL of DMEM maintenance solution containing 2% FBS was added to each well. The cells were placed in an incubator at 37°C and cultured until the required time.

## Determination of cytotoxicity

After the supernatant was collected at different time points of culture, the cell debris was removed by centrifugation at a low temperature of 3000 rpm/min for 20 min, and the supernatant was transferred to a 2 mL EP tube. A lactate dehydrogenase (LDH) assay kit (A020-2–2, Nanjing Jiancheng Bioengineering Institute, China) was used to determine the activity of LDH in the cell culture medium.

## Cellular indirect immunofluorescence assay

MN9D cells were infected with a viral mixture containing JEV (5 MOI). After culture for the indicated time points, the supernatant of the cells was discarded, and the cells were rinsed slightly with PBS. After adding 4% paraformaldehyde at 4°C and fixing at room temperature for 15 min, the cells were washed 3 times with PBS for 5 min each time. The samples were then incubated with PBS (containing 0.3% TritonX-100) was incubated at room temperature for 10 min, followed by immunofluorescence staining.

## Quantitative real-time PCR

Total RNA was extracted from tissues and cells with Trizol reagent (R401-01, Vazyme, China) from tissues and cells, and reverse transcription was performed according to the manufacturer's instructions. Total RNA was reverse transcribed into cDNA via a First-Strand cDNA Synthesis Kit (R223-01, Vazyme, China). The mRNA levels were determined by Real-time fluorescent quantitative polymerase chain reaction (qPCR) using SYBR Green Real-Time PCR Master Mix (Q711-02/03, Vazyme, China) with specific primer pairs (listed in Table 1). The relative RNA levels of specific RNAs were normalized to that of the housekeeping gene ACTB, and the fold change in gene expression was analyzed with the $2^{-\Delta\Delta Ct}$ method. All reactions were performed in triplicate.

**Table 1. qPCR primer sequences.**

| Gene | Forward primers (5'-3') | Downward primers (3'-5') |
|------|-------------------------|--------------------------|
| JEV-E | TGGGACTTTGGCTCTATTGG | AGAACACGAGCACGCCTCCT |
| ACTB | CCTAGGCACCAGGGTGTGAT | AGCACAGGGTGCTCCTCA |
| GSDMD | GCAGAGGCGATCTCATTCCG | CCAAAACACTCCGGTTCTGGTT |
| caspase-1 | GACATCCTTCATCCTCAGAA | CTCCAGCAGCAACTTCAT |
| NLRP3 | AGCCAGAGTGGAATGACACG | GCGCGTTCCTGTCCTTGATA |
| ASC | ACTGTGCTTAGAGACATGGGC | GGTCCACAAAGTGTCCTGTTC |
| IL-1β | GCTTCAGGCAGGCAGTATC | AGGATGGGCTCTTCTTCAAAG |
| IL-18 | ACCAAGTTCTCTTCGTTGAC | TCACAGCCAGTCCTCTTAC |
| TNF-α | TGGCCTCCCTCTCATCAGTT | TTGAGATCCATGCCGTTGGC |
| HMGB1 | AAAGGCTGACAAGGCTCGTT | AAGAAGAAGGCCGAAGGAGG |

## Western blot

Total cellular lysates and tissues were prepared via radioimmunoprecipitation assay buffer (Aidlab Biotechnologies Co., Ltd.) containing protease inhibitors (Aidlab Biotechnologies Co., Ltd.) and phosphataseas (Aidlab Biotechnologies Co., Ltd.). Protein concentrations were determined via a BCA protein assay kit (Aidlab Biotechnologies Co., Ltd). Equal protein quantities were separated by SDS–PAGE and transferred to a polyvinylidene fluoride membrane (Millipore) via a Mini Trans-Blot Cell (Bio-Rad). The blots were probed with the relevant antibodies, and the proteins were detected via enhanced chemiluminescence reagents (Aidlab Biotechnologies Co., Ltd).

## Statistical analysis

ImageJ was applied to integrate and count the immunofluorescence staining results. The trainable Weka Segmentation plug-in of ImageJ was used for machine learning, and each cell was automatically recognized and counted. The luminance of the fluorescence channel where each cell was located was quantified as the mean fluorescence intensity (MFI). Coloc 2 plug-in was used to carry out fluorescence colocalization analysis on the locations of fluorescent molecules in the images formed by overlapping red (IBA-1+, TH+, caspase-1+, GSDMD+ cells) and green (JEV+, TH+ cells) signals. The data results were calculated via regression analysis of scatter plots via Pearson's correlation coefficient (PCC), with PCC values of "1" indicating a perfectly positive correlation, "-1" indicating a perfectly negative correlation, and "0" indicating a random relationship. The statistical analysis software GraphPad Prism was used to calculate the mean and standard error of the mean of each group. The results of each group are represented by the mean ± SEM, and the significance of the mean difference in the means of relevant groups was analyzed by one-way analysis of variance. $P < 0.05$ indicated a significant difference, marked as *; *: $P < 0.05$; **: $P < 0.01$; ***: $P < 0.001$; ****: $P < 0.0001$. Gray level analysis and related calculations were performed on the Western blot results.

## Results

### JEV induced significant pathological injury to dopaminergic neurons

To investigate JEV-induced neuroinflammation, we established an experimental model using 6-week-old female BALB/C mice with intraperitoneal JEV injection, mimicking natural infection. This model was specifically chosen for its < 50% mortality rate, avoiding acute mortality that masks inflammatory processes and enabling focused study on persistent inflammation. Surviving JEV-infected mice exhibited clinical signs from 4-9 days post-infection (dpi), including piloerection, kyphosis, clustering behavior, lethargy, and *in-situ* rotation. Histological analysis (Fig 1A) revealed no significant changes in control groups or at 3 dpi. By 6 dpi, brains showed prominent gliosis, vascular cuffing, and neuronal degeneration, with

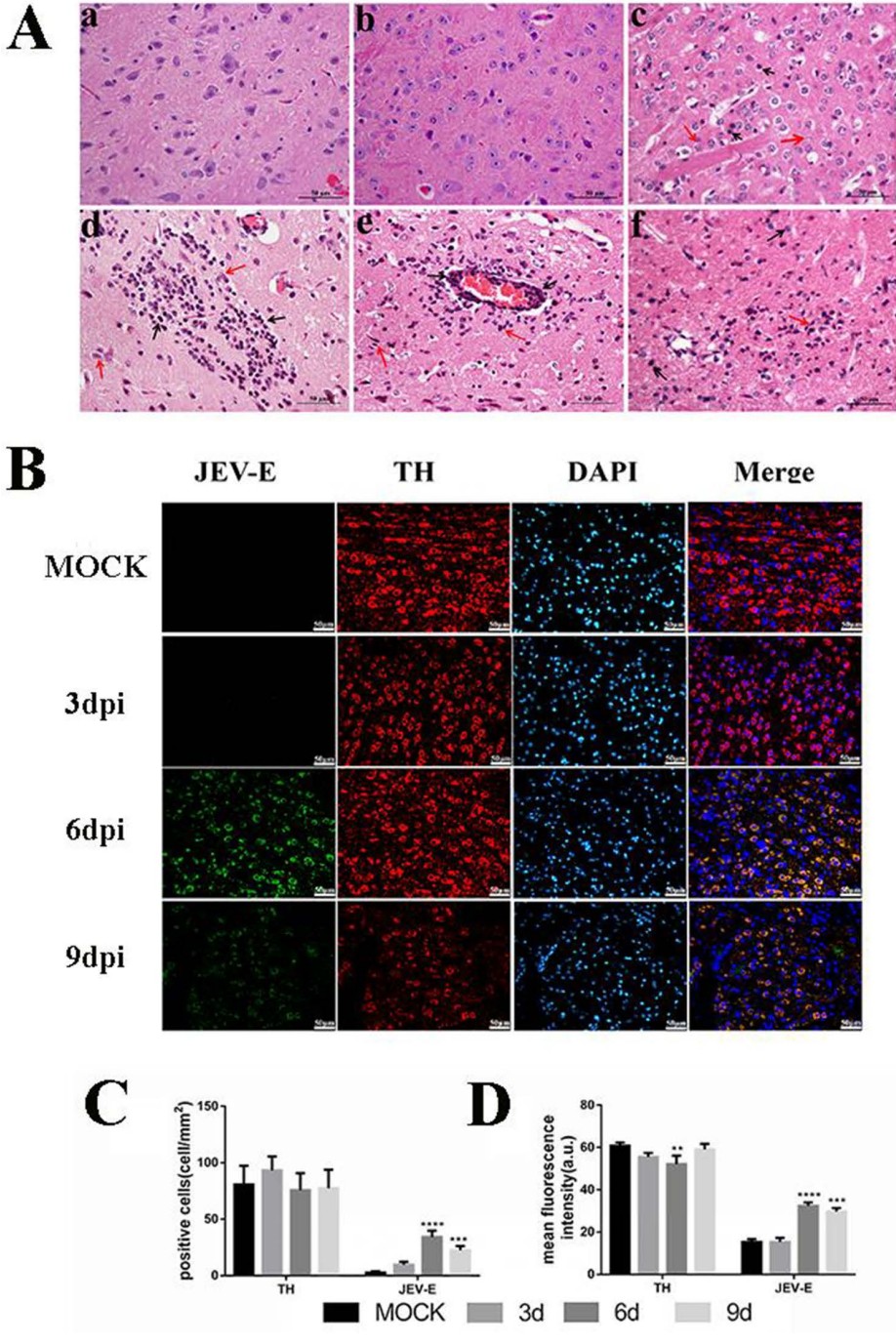

**Fig 1. JEV targets dopaminergic neurons leading to significant pathological damage in mouse brain tissue at 6 and 9 dpi.** A. Pathological tissue changes in BALB/c female mice at 3, 6, and 9 dpi. Mock-infected and JEV-infected mice were observed at each time point. Results showed: a (control group) and b (3 dpi): No significant pathological changes. c (6 dpi): Neuronal degeneration (black arrow) and neuronal necrosis (red arrow). d (6 dpi): Massive proliferation of neuroglial cells (red arrow) and neuronal degeneration (black arrow). e (6 dpi): Vascular cuffing caused by multilayer infiltration of lymphocytes, plasma cells, and monocytes around arterioles and capillaries (black arrow), and massive proliferation of neuroglial cells (black arrow). f (9 dpi): Neuronal degeneration (black arrow) and massive glial cell hyperplasia (red arrow). Scale bar = 50 μm. B. Representative images of co-localization of dopaminergic neurons (TH⁺) and JEV-infected cells (JEV⁺) in mouse brain tissue. Detected by tissue immunofluorescence staining. Quantitative analysis of positive indicators using Image J, presented as: C: Number of positive cells. D: Average fluorescence intensity. Values are expressed as the mean of three biological replicates ± standard error of the mean. **: $P < 0.01$; ***: $P < 0.001$; ****: $P < 0.0001$, vs. mock groups; n = 5.

pathological severity peaking at this time point. Notably, pathological changes attenuated by 9 dpi, likely due to activated glial cells mediating inflammatory resolution. These findings validate the model's utility for investigating JEV-induced neuroinflammation, providing a robust platform for downstream mechanistic studies.

To quantify JEV tropism for dopaminergic neurons, PCC analysis was applied to dual indirect Immunofluorescence assay (IFA) of viral antigen and tyrosine hydroxylase (TH) (Fig 2E). This technique enabled precise spatial mapping of molecular colocalization, distinct from dynamic interaction methods (e.g., Förster-type resonance energy transfer (FRET) with fluorescence cross-correlation spectroscopy (FCCS)) by quantifying fixed-tissue correlations. Pre-infection PCC ≈ 0 confirmed no baseline overlap, while acute infection at 6 dpi demonstrated maximal colocalization (PCC = 0.92), differing significantly from controls (PCC = 0.08) and early infection (PCC = 0.15, $P < 0.001$). By 9 dpi, PCC decreased to 0.54, reflecting reduced viral-neuronal overlap despite persistent infection. These quantitative metrics validated JEV targeting of dopaminergic neurons during acute infection, with subsequent viral spread or cellular damage disrupting spatial association in late-stage disease.

Complementary temporal analysis showed no viral antigen colocalization at 3 dpi, followed by midbrain TH$^+$ neuron infection at 6 dpi and extensive midbrain/diencephalon colocalization at 9 dpi (Fig 1B). Quantification revealed significant increases in JEV$^+$ cell counts at 6 and 9 dpi ($P < 0.05$) (Fig 1C-1D), accompanied by reduced TH$^+$ neuron MFI at 6 dpi ($P < 0.01$) (Fig 1D), indicating impaired dopamine biosynthesis. While absolute TH$^+$ neuron numbers remained unchanged, a non-significant decline suggested potential neurodegeneration. Collectively, PCC analysis provided novel kinetic insights into JEV-neuron interactions, complemented by cell counting and functional assays to establish infection of midbrain/diencephalon dopaminergic neurons.

## Dopaminergic neurons exhibit NLRP3/caspase-1/GSDMD-dependent pyroptosis

Pyroptosis relies on inflammasome activation, with NLRP3 playing a key role in neuroinflammation. Immunohistochemistry revealed NLRP3 and ASC expression in JEV-infected brains. Pre-infection, NLRP3 positivity <50% suggested basal levels. Post-infection, NLRP3 positivity rose to 80% by 9 dpi ($P < 0.01$ vs. control), while ASC increased significantly at 6 dpi ($P < 0.05$) but declined slightly by 9 dpi (Fig 2A and 2B). These dynamics indicate NLRP3 inflammasome activation during JEV-induced neuroinflammation.

Caspase-1 staining in TH$^+$ cells revealed classical pyroptosis pathway activation. Pre-infection and 3 dpi caspase-1$^+$ cells were low, increasing significantly at 6 dpi ($P < 0.05$), with MFI peaking ($P < 0.001$) (Fig 2C and 2D). Caspase-1$^+$ cell counts exceeded TH$^+$ cells due to non-neuronal expression and measurement variability (Fig 2C and 2D). Dopaminergic (DA) neurons and GSDMD were co-stained to assess pyroptosis in JEV-infected brains (Fig 2C). TH$^+$ cell density was highest in mesencephalon/diencephalon, with significantly increased MFI at 6 dpi ($P < 0.05$), likely due to compensatory dopamine secretion. GSDMD$^+$ cells showed robust induction at 6–9 dpi ($P < 0.05$), aligning with NLRP3 inflammasome activation. Pearson's correlation (PCC ≈ 0.5 at 3 dpi, ≈ 1 at 6–9 dpi) confirmed positive colocalization between TH$^+$ and GSDMD$^+$ signals, with GSDMD demonstrating significant differences from the blank group at both 6 dpi and 9 dpi (Fig 2E). The above results indicate that neural cells, represented by dopaminergic neurons, accumulated key executive proteins of pyroptosis before and after JEV infection, suggesting their potential capacity to undergo pyroptosis. Collectively, these findings demonstrate that after viral infection, the NLRP3 inflammasome may be undergoing assembly; with inflammasome activation, this further suggests that pyroptosis may be initiated during this process

Consistent with morphological observations, the transcriptional activation of pyroptosis-associated genes in JEV-infected mice at 6 dpi was further confirmed by RNA-level analysis (Fig 3A). GSDMD expression increased significantly at 6 and 9 dpi ($P < 0.001$), paralleled by caspase-1 upregulation at 6 dpi ($P < 0.05$). NLRP3 and ASC, critical inflammasome components, showed temporal induction: NLRP3 rose from 6–9 dpi ($P < 0.01$), while ASC increased at both time points ($P < 0.05$ and $P < 0.01$). Concomitantly, IL-1β levels peaked at 9 dpi ($P < 0.05$), and IL-18 surged significantly by 9 dpi ($P < 0.001$). HMGB1, a microglial activator, was elevated at 6 dpi ($P < 0.001$). These findings demonstrate JEV

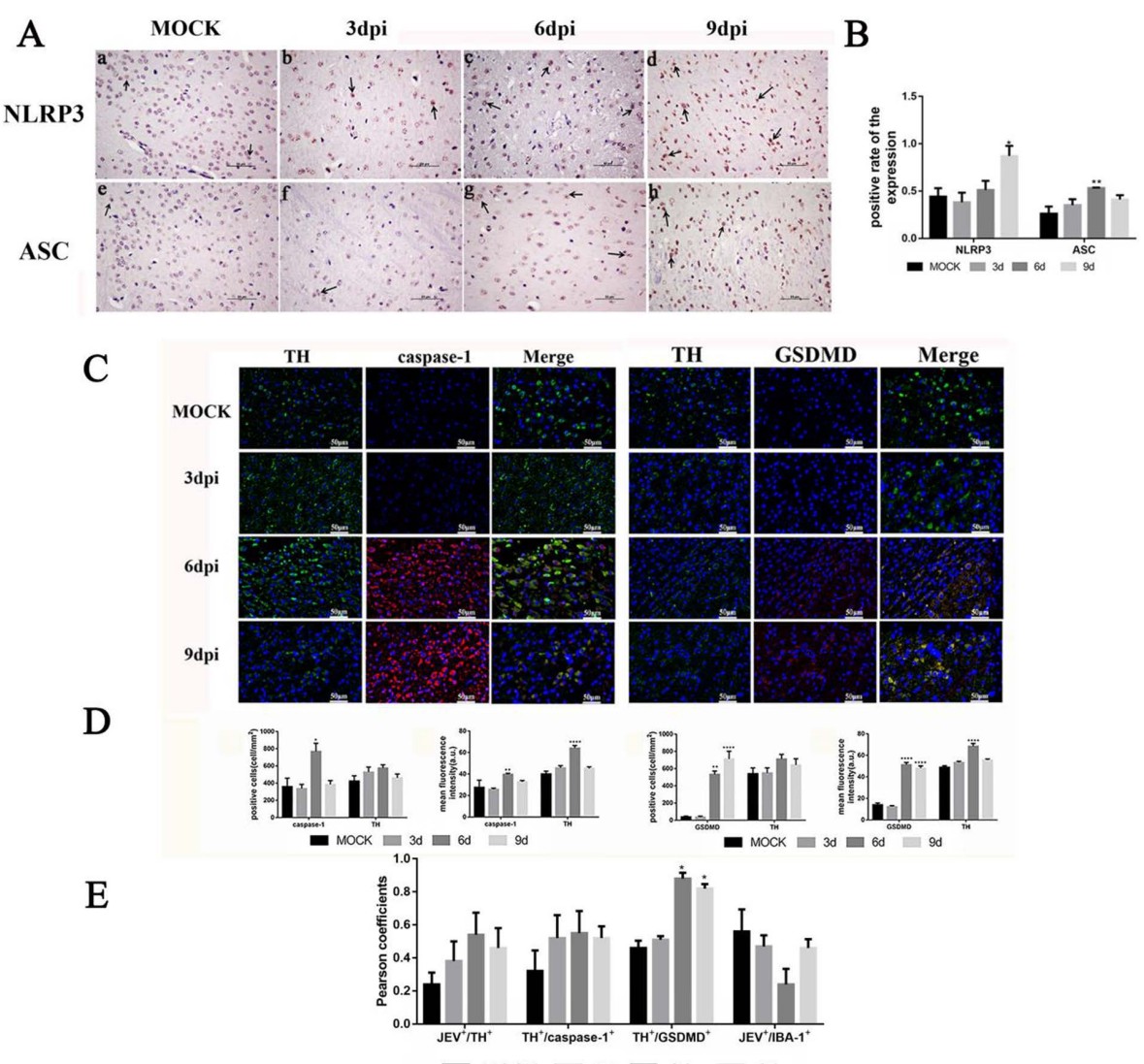

**Fig 2. JEV infection activates NLRP3 inflammasome and caspase-1-GSDMD signaling pathway in dopaminergic neurons of mouse brain tissue.** A. The representative images (a-h) show NLRP3+ and ASC+ cells (black arrows) by IHC: a-d: NLRP3+ staining; e-h: ASC+ staining; a, e (Control): Negative staining; b, f (3 dpi): Sparse positive signals; c, g (6 dpi): Abundant neuronal positive signals; d, h (9 dpi): Widespread cellular positive signals; Scale bar = 50 μm. B. Quantitative analysis of positive cell rates by Image J. Values are expressed as the mean of three biological replicates ± standard error of the mean (n = 5). *: $P < 0.05$, **: $P < 0.01$ vs. mock groups. C. Immunofluorescence co-localization of NLRP3 inflammasome components with dopaminergic neurons. Caspase-1 (red) or GSDMD (red) co-stained with dopaminergic neurons (green). Quantitative analysis by Image J: D. Positive cell and mean fluorescence intensity. E. Pearson's correlation coefficient (PCC) analysis of co-localization. Five random fields per section preprocessed by Image J. One-way analysis of variance results is expressed as mean ± standard error of the mean (*: $P < 0.05$; **: $P < 0.01$. vs. mock groups; n = 5). Representative co-localization images shown in Figs 1B, 2C and 8A.

induces NLRP3/caspase-1/GSDMD-dependent pyroptosis, characterized by sustained inflammation and microglial activation from 6–9 dpi.

Western blot analysis confirmed that JEV infection upregulated the expression of GSDMD, caspase-1, NLRP3, and ASC, consistent with immunohistochemistry results (Fig 3B and 3C). GSDMD expression increased significantly at 6 and 9 dpi ($P < 0.05$ and $P < 0.01$, respectively), while induction of its cleaved form (GSDMD-N) was minimal, potentially due to

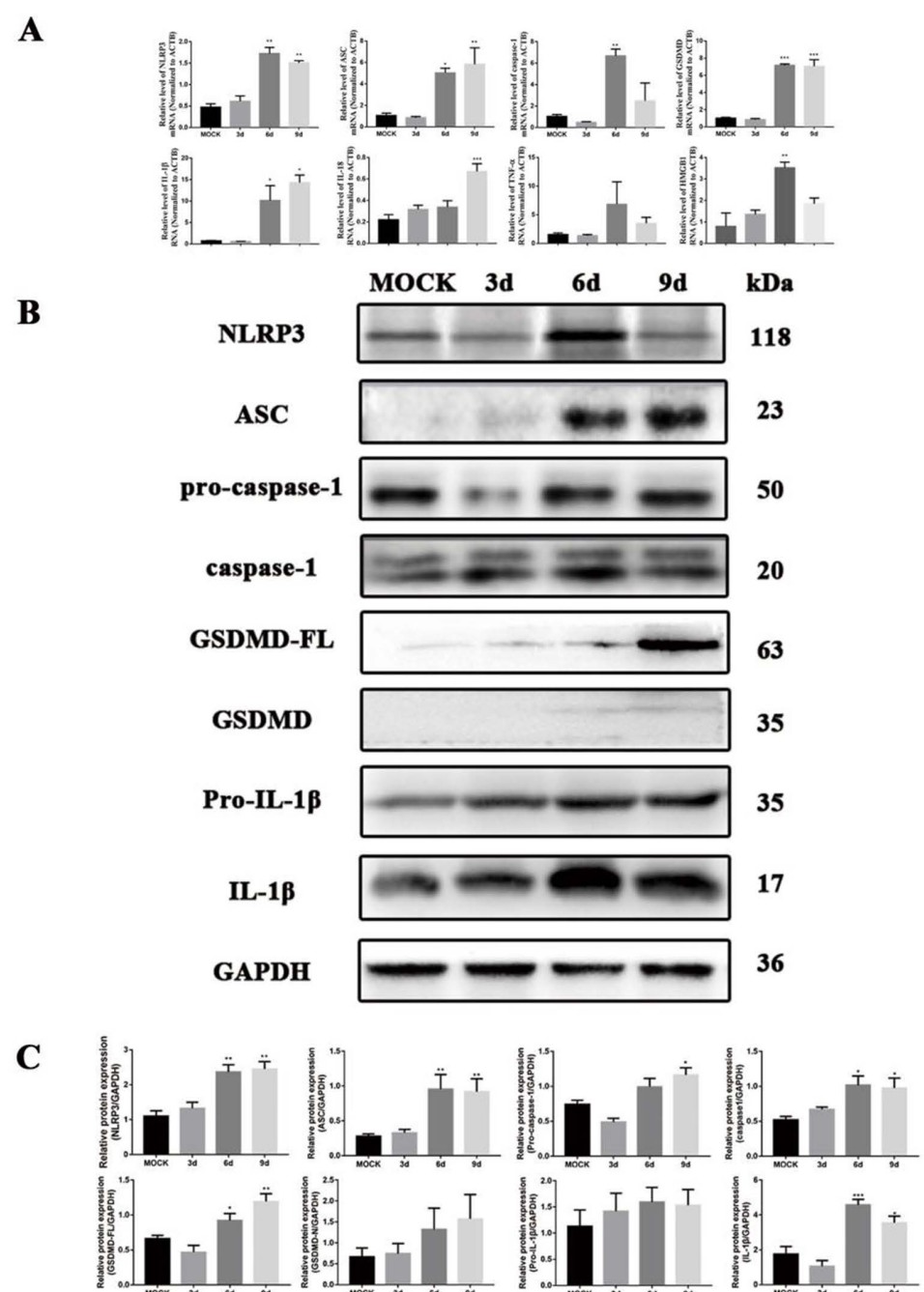

**Fig 3. Infection of mice with JEV induces the activation of the NLRP3/caspase-1/GSDMD pathway at the mRNA level and protein level.** A. qPCR analysis was performed on total RNA isolated from mock-infected and JEV-infected mouse brains at all time points. The mRNA levels of NLRP3, ASC, caspase-1, GSDMD, IL-1β, IL-18, TNF-α and HMGB1 were quantified and normalized against the expression level of ACTB (β-actin); B and C. Western blot analysis detected NLRP3, ASC, Pro-caspase-1, Caspase-1, GSDMD-FL, GSDMD-N, Pro-IL-1β and IL-1β, with protein expression normalized to GAPDH as the internal reference. The values are expressed as the means±SEM. *: $P<0.05$ and **: $P<0.01$ vs. mock group; n=5.

rapid membrane insertion. Pro-caspase-1 levels gradually increased, peaking at 9 dpi ($P<0.05$), accompanied by elevated caspase-1 activity from 6 to 9 dpi ($P<0.05$), indicating inflammasome activation. NLRP3 and ASC exhibited distinct temporal patterns: NLRP3 increased moderately after infection, whereas ASC showed a marked elevation from 6 to 9 dpi ($P<0.01$), underscoring its role in inflammasome assembly. To investigate the inflammatory response triggered by NLRP3 inflammasome activation, we measured the secretion of pro-IL-1β and IL-1β. Pro-IL-1β exhibits basal expression in normal tissues and cells, but its levels progressively increased with prolonged JEV infection, indicating an escalating inflammatory response. Detection of IL-1β cleavage further revealed a significant increase in mature IL-1β at 6–9 dpi ($P<0.05$), peaking at 6 dpi. These results confirm that JEV infection promotes IL-1β secretion, thereby exacerbating neuroinflammation. These results validate that the NLRP3/caspase-1/GSDMD axis is central to JEV-induced pyroptosis. Collectively, these findings establish a causal link between JEV infection, NLRP3 inflammasome activation, and the execution of pyroptosis, highlighting the axis as a potential therapeutic target for mitigating JEV-induced neuroinflammation.

### JEV induces pyroptosis in MN9D cells through the NLRP3 inflammasome

The MN9D cell line, derived from mouse meso-tegmental dopaminergic neurons fused with N18TG2 neuroblastoma cells, retains dopaminergic characteristics (e.g., tyrosine hydroxylase expression, dopamine synthesis/release) and is widely used in dopaminergic injury models. In JEV-infected MN9D cells observed via light microscopy, morphological changes emerged at 24 hpi (Fig 4A): control cells showed elongated, regular spindles, while infected cells displayed shrinkage, rounding, irregular shapes, and increased secretions. Late-stage infection caused membrane damage, cell fragmentation, and floating debris. These results model JEV-induced dopaminergic neuron pathology. LDH release was measured to assess membrane rupture in JEV-infected MN9D cells (Fig 4B). LDH levels increased progressively post-infection, rising significantly at 48–60 hpi ($P<0.01$), indicating membrane permeability changes consistent with pyroptotic cell death. This correlated with observed morphological damage and suggests inflammatory factor release during late-stage infection.

qPCR analysis revealed dynamic viral replication kinetics in dopaminergic neurons (Fig 4C): JEV load increased exponentially ($P<0.0001$), peaking at 24 hpi before declining slightly. This biphasic pattern reflects initial intracellular viral amplification followed by cell lysis and virion release into the supernatant. Concomitant western blot analysis (Fig 6B and 6C) confirmed viral entry and replication via envelope (E) protein expression, which rose progressively post-infection with significant increases at 36–60 hpi ($P<0.05$). IFA validated JEV tropism for dopaminergic neurons, showing cytoplasmic viral signals at 24 hpi (Fig 4D). Infected cell numbers increased exponentially until 48 hpi before plateauing, indicating active viral spread. Late-stage infection (72 hpi) demonstrated intensified peripheral signals likely due to pyroptosis-induced membrane permeability changes, complicating signal interpretation (Fig 4). Collectively, these data characterize JEV replication kinetics and cytopathic effects in dopaminergic neurons, highlighting early intracellular amplification followed by extracellular dissemination.

To confirm the NLRP3 inflammasome-mediated pyroptosis in JEV-infected MN9D cells, we investigated the expression dynamics of NLRP3, ASC, caspase-1, and GSDMD (Fig 5). Basal expression levels of all these components were relatively low prior to infection. Following JEV infection, their expressions exhibited an orderly temporal pattern. Specifically, NLRP3 expression was significantly upregulated from 24 to 48 hours post-infection (hpi) but subsequently declined at 60 hpi, which may be attributed to protein degradation associated with late-stage cell death. Concomitantly, upon sustained activation signals from NLRP3, ASC expression was robustly induced between 48 and 60 hpi, indicating that the inflammasome assembly scaffold was fully established at this stage. Notably, since the antibodies used in this study target the full-length, uncleaved forms of these proteins, the upregulation of caspase-1 and GSDMD detected essentially reflects the synthesis and accumulation of their full-length precursors. This accumulation precedes the significant induction of ASC, suggesting that upon sensing viral insults, MN9D cells prioritize the biosynthesis of pyroptotic execution molecules to lay the foundation for the subsequent rapid pyroptotic response. In summary, these dynamic expression profiles indicate that NLRP3 inflammasome activation reaches its peak at approximately 48 hpi, which may subsequently trigger caspase-1-dependent GSDMD cleavage and the

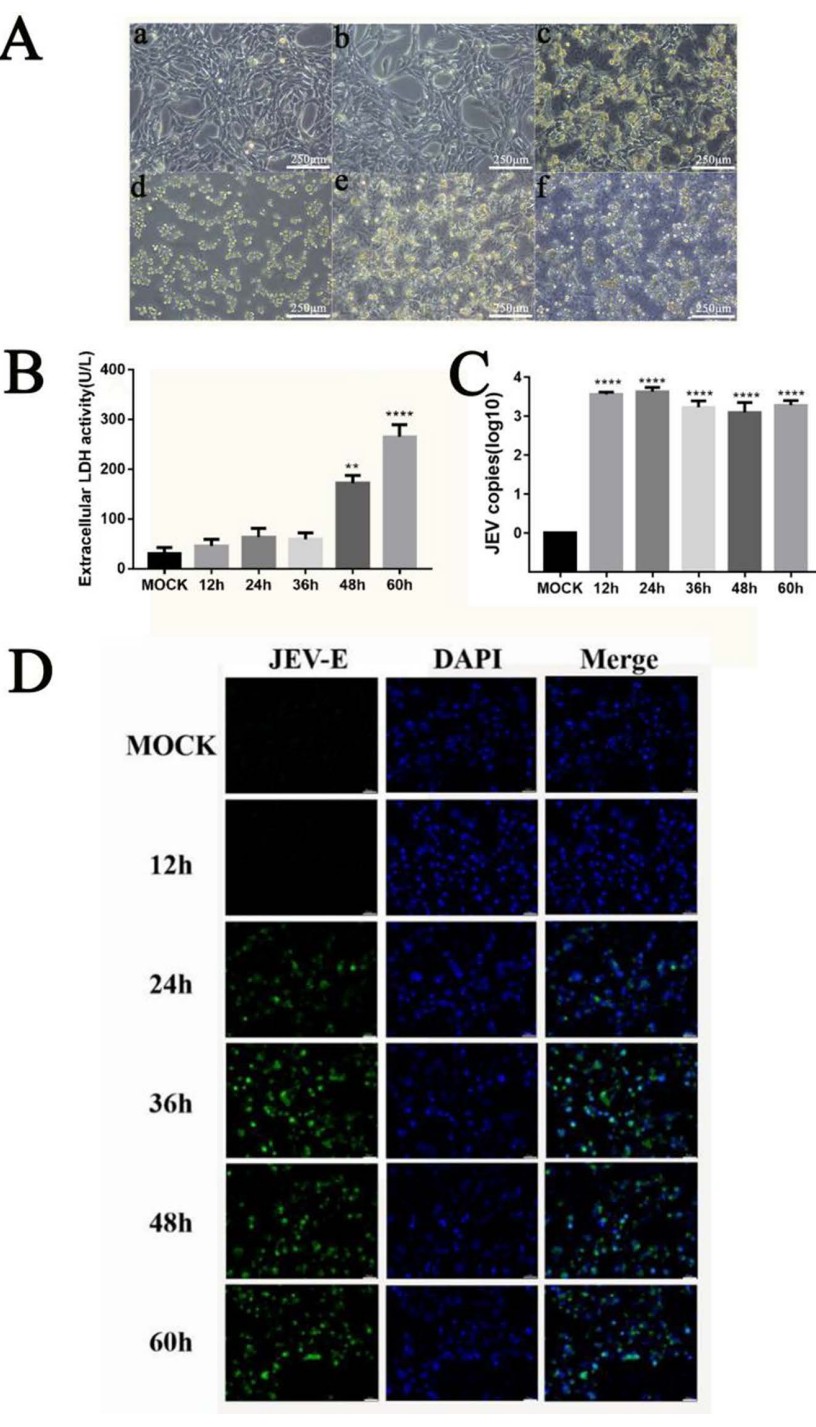

**Fig 4. The dopaminergic neuron cell line MN9D infects JEV *in vitro*, causing changes in cell morphology and a significant expansion of the virus.** A. Representative images of the morphological changes in MN9D cells before and after JEV infection. a. In the control group, the cell morphology was regular; b. At 12 hpi, the cells were arranged in a regular manner with no obvious lesions; c. At 24 hpi, the cells shrank and became round with irregular shapes; d. At 36 hpi, adherent cells were significantly reduced, and the morphology continued to be irregular; e. At 48 hpi, the cells burst, and a large amount of cell secretions appeared; f. At 60 hpi, large amounts of cellular debris and severe cellular vacuolization occurred. Scale bar = 250 μm. B. The integrity of the cell membrane was assessed via a lactate dehydrogenase (LDH) activity assay. The results demonstrated that the cell membrane of MN9D cells was significantly damaged from 48 to 60 hpi. C. Viral load of MN9D cells at different times after infection with JEV (the values are expressed

as the mean±SEM. ****: $P<0.0001$ vs. mock group). D. Immunofluorescence was performed to visualize the infectivity of the virus within the cells. The results indicated that the virus showed obvious proliferation as early as 12 hpi. The virus was marked with the JEV-E (green) marker, and the nucleus was stained with DAPI (blue). Scale bar = 10 μm;.

onset of pyroptosis. This process is predominantly confined to the middle and late phases of JEV infection, and the precise molecular mechanisms governing this cascade remain to be further elucidated.

Investigating transcriptional mechanisms of JEV-induced pyroptosis in dopaminergic neurons, qPCR was used to analyze pyroptosis-related cytokines and inflammatory factors (Fig 6A). NLRP3 and ASC mRNA levels increased significantly at 48–60 hpi ($P<0.05$), coinciding with GSDMD upregulation from 36–60 hpi ($P<0.01$) and caspase-1 activation at 60 hpi ($P<0.05$). IL-1β showed transient early suppression followed by robust induction at 24–60 hpi ($P<0.05$), consistent with pyroptotic IL-1β secretion. IL-18 and TNF-α exhibited biphasic responses: IL-18 decreased at 24–36 hpi ($P<0.01$) but surged at 60 hpi ($P<0.0001$), while TNF-α declined at 12–48 hpi ($P<0.05$) before rising at 60 hpi ($P<0.01$). These dynamics suggest stage-specific immune regulation during infection. Collectively, transcriptional upregulation of NLRP3 inflammasome components (NLRP3/ASC), caspase-1, and GSDMD supports JEV-induced pyroptosis in late-stage infection, with IL-1β/IL-18/TNF-α contributing to sustained neuroinflammation.

Determining direct JEV-induced activation of caspase-1 and GSDMD, Western blot analysis examined NLRP3 inflammasome components during infection (Fig 6B-6C). NLRP3 showed dynamic fluctuations: pre-infection expression decreased significantly at 12–24 hpi ($P<0.05$), recovered at 36 hpi, declined again at 48 hpi ($P<0.05$), and surged at 60 hpi ($P<0.05$). ASC followed a distinct pattern: low pre-infection levels rose progressively post-infection, with significant increases at 36 hpi ($P<0.001$) and 60 hpi ($P<0.01$), despite a dip at 24 hpi. Caspase-1 cleavage, undetectable pre-infection, increased significantly from 36–60 hpi ($P<0.01$), paralleling GSDMD processing into its active N-terminal form (GSDMD-N). GSDMD expression steadily rose from 36–60 hpi ($P<0.001$ vs. control), with GSDMD-N peaking at 48–60 hpi ($P<0.05$ and $P<0.01$, respectively). A subpopulation of "rapid responder" cells drives the initial response (12 hours), followed by a delayed response (36 hours) dominated by cells with slower infection progression. These kinetics suggest NLRP3 inflammasome activation at 36–60 hpi drives caspase-1-mediated GSDMD cleavage, promoting pyroptosis. The biphasic NLRP3 response likely reflects inflammasome assembly/disassembly cycles, while ASC sustained upregulation underscores its role in inflammasome stabilization during late-stage infection. To explore the role of NLRP3 inflammasome in inflammatory regulation, we analyzed the expression kinetics of pro-IL-1β and its mature form. Results show that pro-IL-1β expression peaked at 36 hpi, followed by a transient downregulation at 48 hpi. Concomitantly, mature IL-1β exhibited an inverse expression pattern: a significant increase after pro-IL-1β cleavage, followed by a decline, which validated the canonical pro-IL-1β maturation pathway during inflammation. Notably, IL-1β expression showed a transient reduction at 24 hpi, likely due to cellular negative feedback regulation. However, persistent viral replication and inflammatory cascade amplification drove a sustained IL-1β upregulation, forming a self-reinforcing injury-inflammation cycle. Although IL-1β levels dipped slightly at 60 hpi, they remained significantly elevated compared to basal levels ($P<0.05$). These data reveal a complex scenario: the virus undergoes multiple rounds of replication and spread within the cell population, with each round triggering a robust innate immune response that subsequently subsides as responding cells undergo pyroptosis. These findings illustrate the dynamic interplay between viral infection and inflammatory signaling, highlighting the NLRP3 inflammasome-IL-1β axis as a key regulator of JEV-induced neuroinflammation.

## MCC950 inhibits the activation of NLRP3 inflammasome without preventing viral infection

Preclinical studies have established that JEV potently activates the NLRP3 inflammasome, prompting us to investigate whether NLRP3 inhibition by MCC950 could mitigate JEV-induced neuroinflammation [14,15]. Mice received daily MCC950 injections 3 days prior to JEV infection and until 9 dpi to achieve steady-state drug concentrations (Fig 7A). All

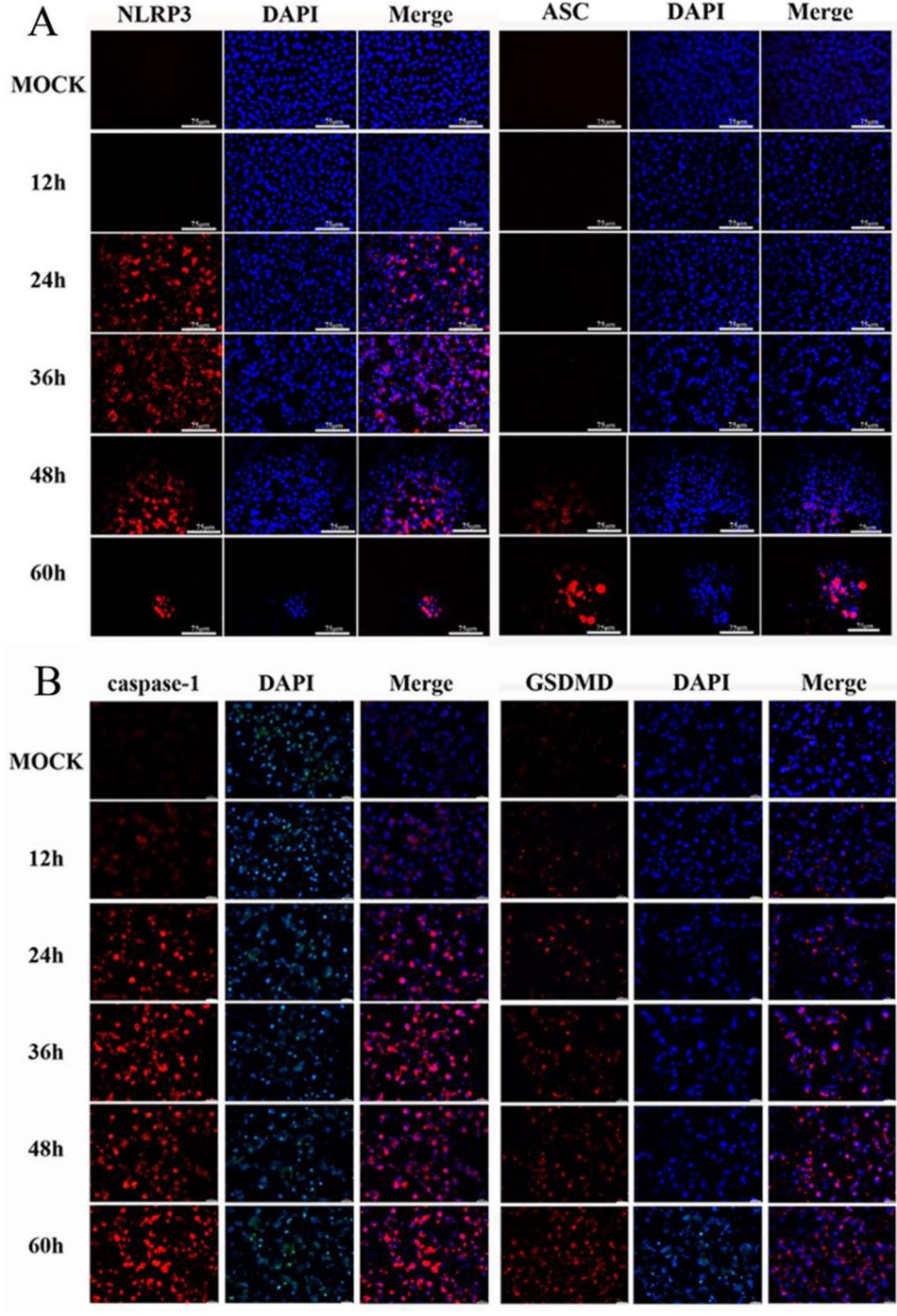

**Fig 5. JEV infection triggers NLRP3 inflammasome activation and pyroptosis in MN9D cells.** A. Representative immunofluorescence images illustrating the expression patterns of NLRP3 (red) and ASC (red). The results demonstrated that NLRP3 started to be significantly activated at 24 hours post-infection, while ASC showed obvious activation at 48 hours post-infection. Scale bar = 75 μm. B. Representative immunofluorescence images showing the expression patterns of caspase-1 (red) and GSDMD (red). The results indicated that caspase-1 began to be significantly activated at 24 hours post-infection, whereas GSDMD exhibited notable activation as early as 12 hours post-infection. Scale bar = 10 μm.

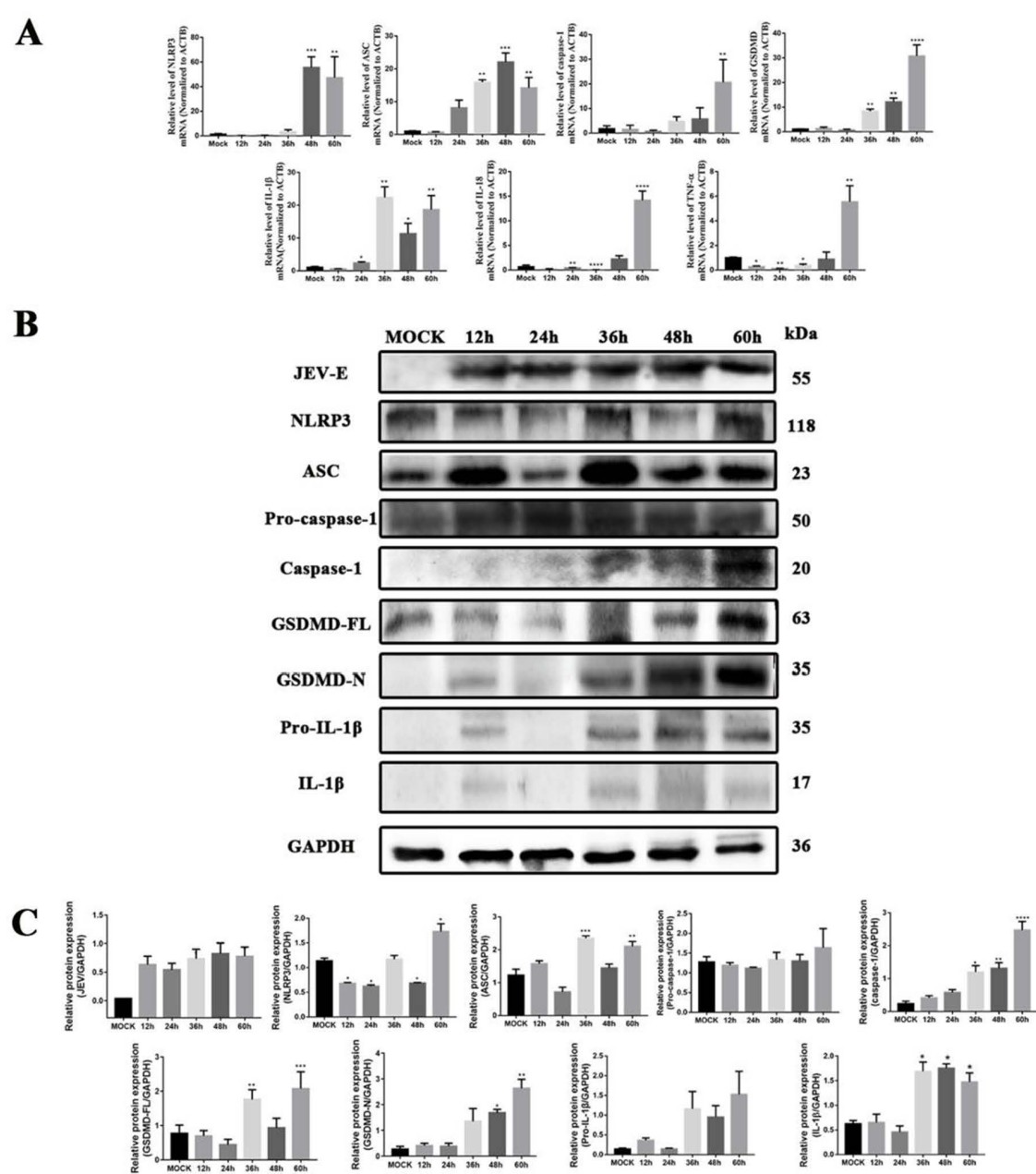

**Fig 6. JEV activates the NLRP3 inflammasome in MN9D cells *in vitro*.** A. qPCR analysis was performed on total RNA isolated from mock-infected and JEV-infected MN9D cells at all time points; the mRNA levels of NLRP3, ASC, caspase-1, GSDMD, IL-1β, IL-18 and TNF-α were quantified and normalized against the expression level of ACTB; B and C. Western blot analysis detected NLRP3, ASC, Pro-caspase-1, Caspase-1, GSDMD-FL, GSDMD-N, Pro-IL-1β and IL-1β, with protein expression normalized to GAPDH as the internal reference. The values are expressed as the mean ± SEM. *: $P < 0.05$, **: $P < 0.01$, ***: $P < 0.001$, and ****: $P < 0.0001$ vs. the mock group.

mice survived the experimental period, enabling comprehensive clinical scoring of MCC950 efficacy. Results showed that MCC950-treated JEV-infected mice developed typical neurological symptoms by 6 dpi, including kyphosis, lethargy, agitation, and tremors. Strikingly, these symptoms did not differ significantly from those in JEV-only controls across the observation period, indicating that NLRP3 inhibition failed to ameliorate JEV-induced clinical manifestations (Fig 7B).

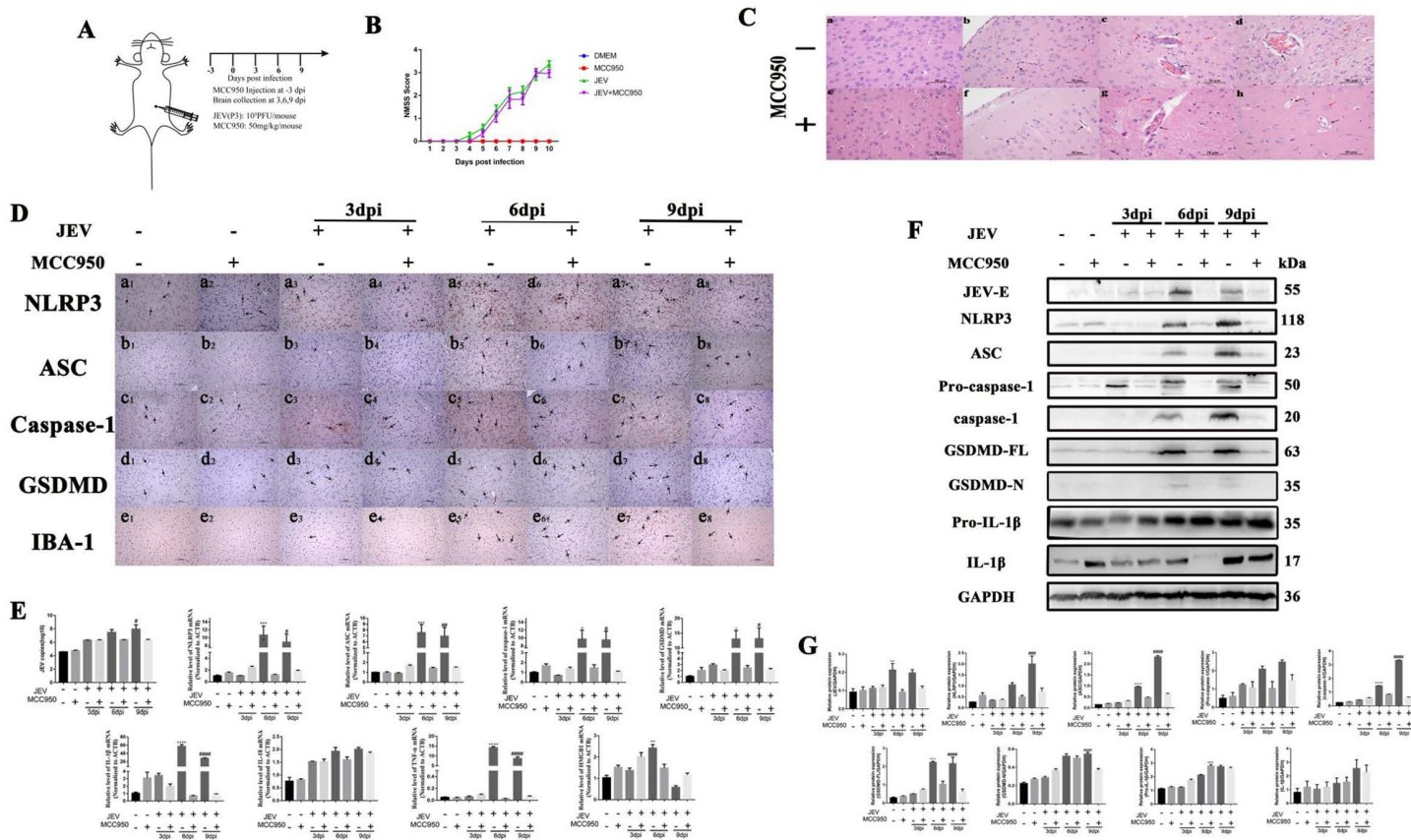

**Fig 7. MCC950 mitigates but cannot prevent inflammation in JEV-infected mice *in vivo*.** A. Schematic representation of viral infection and MCC950 intervention; B. Uninterrupted Mouse Survival Enables MCC950 Efficacy Evaluation via Clinical Scoring. C. HE-stained brain sections from mice 3 dpi, 6 dpi and 9 dpi with JEV and treated with MCC950 or the control DMEM. The images shown are representative of five mice for each condition. a. Control group. b. MCC950. c and d. Microvascular proliferation (dark arrow) after infection with JEV or JEV⁺ MCC950 at 3 dpi. e and f. Vascular cuffs (dark arrows) and degeneration of neurons (red arrows) after infection with JEV or JEV + MCC950 at 6 dpi. g. Vascular cuffs (dark arrows) and degeneration of neurons (red arrows) after infection with JEV at 9 dpi. h. Microvascular proliferation (dark arrow) and degeneration of neurons (red arrows) after infection of JEV + MCC950 at 9 dpi. Scale bars = 50 μm; D. IHC showing NLRP3+ (a1-a8), ASC+ (b1-b8), caspase-1+ (c1-c8), GSDMD+ (d1-d8) and IBA-1+ (e1-e8) (dark arrows). Scale bars = 100 μm; E. qPCR analysis of total RNA from mock- and JEV-infected mouse brains at all time points quantified JEV viral load (mean ± SEM; *: $P < 0.05$ vs. mock) and mRNA levels of NLRP3, ASC, caspase-1, GSDMD, IL-1β, IL-18, TNF-α, HMGB1, normalized to reference gene ACTB; F and G. Western blot analysis detected NLRP3, ASC, Pro-caspase-1, Caspase-1, GSDMD-FL, GSDMD-N, Pro-IL-1β and IL-1β, with protein expression normalized to GAPDH as the internal reference. The values are expressed as the mean ± SEM; * : $P < 0.05$; ** : $P < 0.01$; *** : $P < 0.001$; **** : $P < 0.0001$ vs. 6 dpi with JEV group; #: $P < 0.05$; ##: $P < 0.01$; ###: $P < 0.001$; ####: $P < 0.0001$; vs. 9 dpi with JEV group; n = 5.

Histopathology analysis (Fig 7C) revealed persistent vascular proliferation, neuronal degeneration, and lymphocytic infiltration, indicating unmitigated neuroinflammation.

To confirm whether MCC950 exerts inhibitory effects on key components of the NLRP3 inflammasome in brain tissues, we used IHC to observe the positive signals of NLRP3, ASC, caspase-1, and GSDMD (Fig 7D). The results showed that NLRP3 was expressed not only weakly in neurons but also in glial cells, which posed challenges for quantitative detection. This finding also suggests that while MCC950 can reduce the expression of NLRP3 in neurons, the activation of NLRP3 in glial cells appears to be independent of the presence of MCC950. In contrast, the positive signals of downstream signaling molecules ASC, caspase-1, and GSDMD were significantly reduced in both neurons and glial cells, indicating that MCC950 alleviates pyroptosis to a certain extent.

qPCR and Western blot (Fig 7E-7G) confirmed MCC950 reduced NLRP3/ASC/caspase-1/GSDMD mRNA and protein levels at 6–9 dpi ($P < 0.05$), but failed to resolve clinical symptoms, suggesting incomplete reversal of established neuronal injury. MCC950 transiently reduced viral titers and E protein expression at 6 dpi ($P < 0.01$) but not thereafter. Transcriptional analysis showed selective suppression of NLRP3-dependent cytokines (IL-1β and TNF-α; $P < 0.0001$), while IL-18 remained unchanged and HMGB1 increased at 9 dpi, reflecting NLRP3-independent inflammatory pathways. We evaluated IL-1β protein expression and found that MCC950 treatment significantly increased Pro-IL-1β levels at 6 dpi compared to JEV-infected controls ($P < 0.05$). No significant differences in Pro-IL-1β expression were observed at other time points between MCC950-treated and JEV-only groups. This temporal pattern constructs a dynamic model of "early inhibition-late escape", which combined with elevated HMGB1 mRNA at 9 dpi, infers a mechanistic hypothesis that "late-stage viral replication triggers NLRP3-independent inflammation". IHC revealed unaltered microglial activation (IBA-1 staining) in MCC950-treated mice, indicating viral clearance occurs independently of NLRP3. This suggests microglia drive inflammation via NLRP3-independent mechanisms despite MCC950 intervention. These findings highlight a dual role of MCC950: while it suppresses NLRP3 inflammasome activation, unregulated microglial activation via NLRP3-independent pathways (e.g., HMGB1-mediated phagocytosis) sustains neuroinflammation. The therapeutic limitation of isolated NLRP3 inhibition underscores the need for combinatorial strategies targeting both inflammasome-dependent and microglia-driven inflammatory networks in JEV infection.

### The impact of the NLRP3/caspase-1/GSDMD signaling pathway on microglial activation

Microglia play a central role in JEV-induced neuroinflammation, yet in vivo studies with MCC950 reveal a critical insight: microglial activation is uncoupled from NLRP3 inflammasome activation, challenging the hypothesized cascade of microglia-NLRP3-neuronal pyroptosis. This dissociation implies alternative inflammatory mechanisms requiring further investigation.

Dual immunofluorescence staining in JEV-infected mice (Fig 8A) shows that uninfected microglia (IBA-1+) exhibit a quiescent, rounded morphology. By 3 dpi, microglia numbers increase without significant morphological change or viral antigen detection. At 6 dpi, numerous JEV+ cells emerge alongside activated microglia with elongated processes aggregating around infected cells. By 9 dpi, virus-infected cells are surrounded by highly activated microglia with extended processes. Quantitative analysis (Fig 8B) confirms significant increases in JEV+ cells and microglial activation ($P < 0.05$) from 6-9 dpi, with peak activation at 6 dpi ($P < 0.0001$), establishing microglial activation as a hallmark of neuroinflammation. PCC analysis (Fig 2E) reveals a weak random correlation ($0 < PCC < 0.5$) between JEV+ cells and IBA-1+ microglia, indicating JEV infects microglia sporadically while preferentially targeting other cell types. The non-overlapping fluorescence signals of IBA-1+ (red) and JEV+ (green) cells further support this. Collectively, these findings demonstrate that JEV induces neuroinflammation through both infectious and noninfectious pathways, with the latter potentially exacerbating neuronal damage independently of NLRP3 inflammasome activation in microglia.

To investigate the effects of JEV infection on microglial function, the BV2, mouse microglial cell line was selected for infection experiments. Morphological observations indicated that prior to infection (Fig 8C), the microglia predominantly exhibited a round or oval shape, with few cells displaying branching. At 12 and 24 hpi, the cells were evenly distributed and mostly round, although some began to show branching changes and aggregation. By 36 hpi, the cells exhibited elongated spindle shapes and pronounced branching alterations. Over time, significant morphological changes were observed, with most cells exhibiting an activated state characterized by enlarged cell bodies and flattened shapes (Fig 8C). Additionally, the number of cells decreased significantly, suggesting that microglia become markedly activated as the duration of JEV infection increases.

Previous studies have shown that JEV-infected microglia undergo apoptosis, but the role of the NLRP3 inflammasome in modulating microglial function and pyroptosis remains unclear. To address this, we investigated pyroptosis-related factor expression during JEV infection (Fig 8D). Viral replication, as measured by JEV-E protein levels, significantly increased at 12 hpi and remained elevated ($P < 0.01$). For NLRP3 inflammasome components, NLRP3 mRNA showed

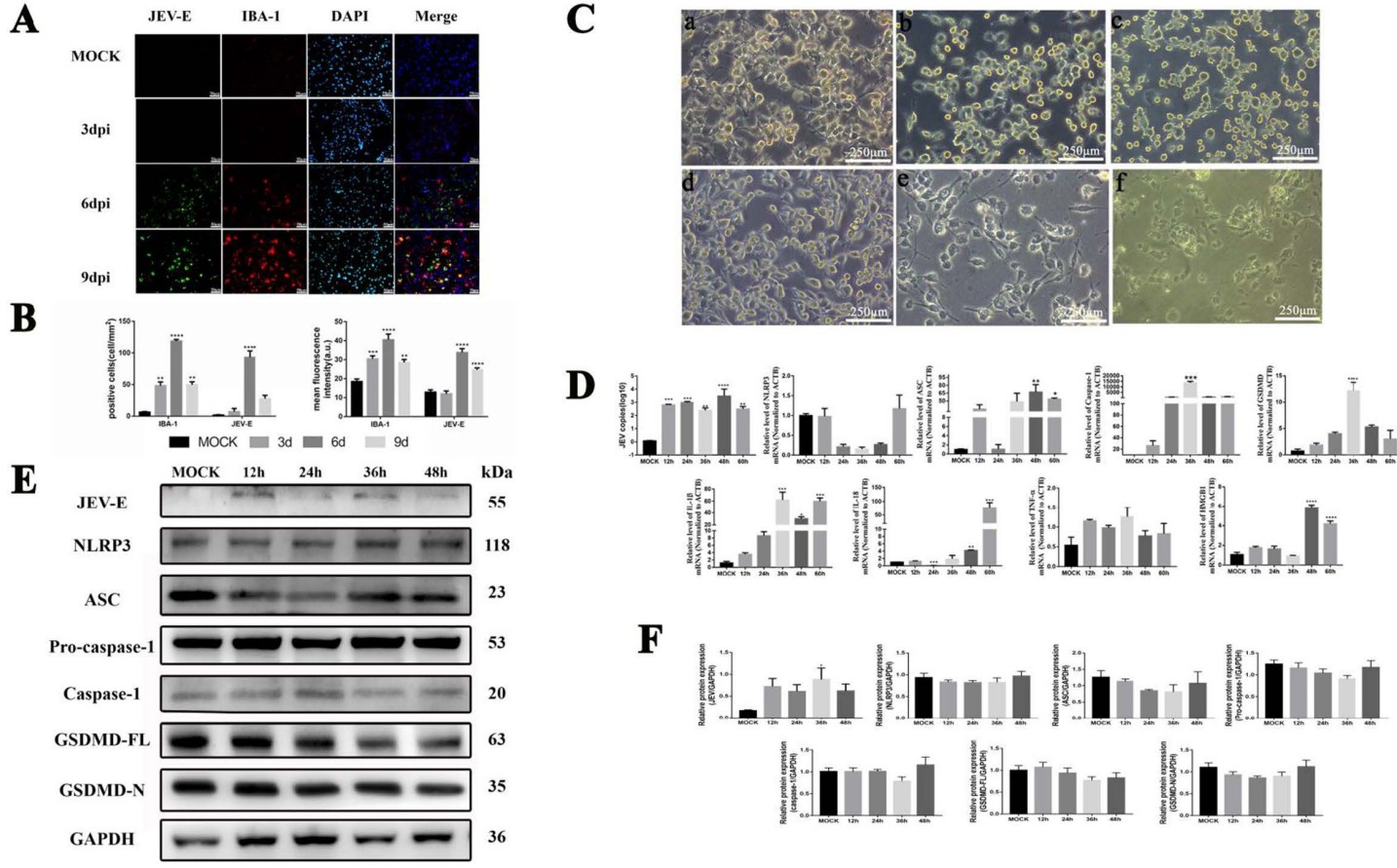

**Fig 8. Changes of microglia before and after JEV infection *in vitro* and *in vivo*.** A. Representative photomicrographs illustrating the immunoreactivity of IBA-1 (red) and JEV-E (green) *in vivo*. Scale bar: 50 μm; B. Quantification of positive cells and the mean fluorescence intensity of IBA-1 and JEV-E (mean±SEM; **: $P<0.01$; ***: $P<0.001$; ****: $P<0.0001$, vs. mock groups; n=5). C. Morphological changes in BV2 cells before and after JEV infection *in vitro*. a. Control group; b. At 12 hpi, the cells appeared round or oval with regular arrangement; c. At 24 hpi, the cells began to aggregate into clumps with irregular morphology; d. At 36 hpi, the cells were predominantly spindle shaped, and aggregation increased; e. At 48 hpi, numerous branching cells emerged, with some cells becoming larger; f. At 60 hpi, the number of cells decreased, resulting in a more branched and clustered appearance; D. Cytokines related to the classical pyroptosis pathway and other inflammatory cytokines in BV2 cells before and after JEV infection were detected via qPCR. Total RNA from mock- and JEV-infected BV2 cells at all time points was analyzed to quantify mRNA levels of GSDMD, caspase-1, NLRP3, ASC, IL-1β, IL-18 and TNF-α, normalized to reference gene ACTB (mean±SEM; *: $P<0.05$; **: $P<0.01$; ***: $P<0.001$; ****: $P<0.0001$; vs. mock group). E-F. Western blot analysis detected NLRP3, ASC, Pro-caspase-1, Caspase-1, GSDMD-FL, GSDMD-N, Pro-IL-1β and IL-1β, with protein expression normalized to GAPDH as the internal reference. The values are expressed as the mean±SEM. *: $P<0.05$, **: $P<0.01$ and ***: $P<0.001$ vs. mock group.

mild fluctuations without significance, while ASC mRNA increased at 12 hpi, decreased at 24 hpi, and sustained high expression from 36–60 hpi ($P<0.05$). Caspase-1 mRNA progressively rose, peaking at 36 hpi ($P<0.001$), consistent with ASC-driven activation, and GSDMD mRNA increased from 12–36 hpi ($P<0.0001$) before declining but staying above control levels. Inflammatory cytokine analysis revealed IL-1β, an NLRP3-dependent cytokine, significantly increased at 36–60 hpi ($P<0.05$), while IL-18 showed biphasic regulation—decreasing at 24 hpi ($P<0.001$) and rebounding at 48–60 hpi ($P<0.01$). HMGB1 increased significantly at 48–60 hpi ($P<0.0001$), whereas TNF-α trended upward without significance.

In the detection of relevant protein levels, we found that microglia exhibited relatively high basal expression of the NLRP3 inflammasome and its effector proteins under uninfected conditions. Following JEV infection, the expression levels of NLRP3, ASC, caspase-1, and GSDMD decreased to varying degrees, but these reductions were not statistically

significant. A slight upward trend was observed at 48 hours post-infection, which also lacked statistical significance. These results indicate that JEV infection induces changes in the NLRP3/ASC/GSDMD signaling pathway in microglia, but such changes do not trigger membrane perforation, a hallmark of canonical pyroptosis, suggesting that microglia do not undergo significant pyroptosis.

Collectively, these findings demonstrate that the NLRP3 inflammasome is initiated at 12 hours post-infection, with subsequent activation of the caspase-1/GSDMD cascade at 48 hours post-infection. However, the absence of GSDMD-N indicates the absence of canonical pyroptosis (Fig 8E and 8F). In vivo MCC950 inhibition experiments further confirmed that NLRP3/caspase-1/GSDMD-mediated pyroptosis is not the primary mechanism underlying microglial damage, as delaying pyroptosis failed to attenuate neuroinflammation. These results highlight the complexity of the interaction between JEV and microglia, wherein NLRP3 activation contributes to the inflammatory response without promoting canonical pyroptosis.

## Discussion

Pyroptosis, a form of programmed cell death mediated by caspase-1-dependent GSDMD cleavage, has been mechanistically linked to inflammatory pathogenesis in viral encephalitis [16]. In murine models of JEV infection, GSDMD upregulation coincides with clinical symptom onset, suggesting a causal role for pyroptosis in disease progression [17]. This inflammatory cell death pathway is activated by viral components engaging pattern recognition receptors, such as the NLRP3 inflammasome, leading to caspase-1 maturation and subsequent IL-1β secretion. Notably, NLRP3 inflammasome activation occurs via classical (caspase-1-dependent) or nonclassical (mouse caspase-11-dependent) pathways. Despite robust inflammasome activation, IL-18 transcription levels paradoxically failed to correlate with caspase-11-mediated cleavage [18]. This discrepancy indicates the nonclassical pathway is not functionally engaged during JEV infection, likely due to species-specific caspase-11 inefficiency in processing mouse Pro-IL-18. Consequently, pyroptosis in JEV-infected cells is proposed to proceed exclusively through the classical caspase-1-dependent axis, obviating the need for caspase-11-specific measurements.

Neurotropic viruses like JEV establish tropism for specific neuronal populations, thereby disrupting neurotransmission and initiating pathological cascades [19,20]. In the thalamostriate circuit, JEV preferentially infects DA neurons, as confirmed by double-immunofluorescence labeling demonstrating colocalization of viral antigens with TH. This tropism results in significant reductions in TH immunoreactivity and DA neuron density, directly contributing to motor deficits observed in infected hosts. The flaviviruses evade host immunity through complex mechanisms, complicating the development of antiviral therapeutics [21]. However, modulating key inflammatory pathways—such as pyroptosis—may mitigate disease severity. For example, caspase-1 inhibitors reduce ZIKV-induced neural precursor cell death [22], while Dengue virus (DENV-1) infection triggers macrophage pyroptosis via NLRP3/caspase-1/GSDMD signaling, a process attenuated by caspase-1 inhibitor Ac-YVAD-cmk [23]. Similarly, traditional Chinese medicine formulations like Xijiao Dihuang decoction suppress influenza virus-induced pyroptosis by targeting the ROS/NLRP3 axis [24]. These findings highlight pyroptosis as a conserved inflammatory response across viral infections, though its role in JEV pathogenesis remains incompletely understood.

To explore the role of NLRP3 inflammasome in JEV-induced neuroinflammation, we administered MCC950, a selective NLRP3 inhibitor [25,26], to infected mice. Therapeutic rationale for NLRP3 inhibitors lies in their "upstream specificity": precise blockade of NLRP3-mediated pathology with minimal interference to other immune pathways [27]. Downstream targets (ASC/caspase-1/GSDMD) pose higher risks of off-target effects due to their involvement in multiple inflammatory pathways. Pharmacological inhibition offers advantages over gene knockout by avoiding compensatory responses (e.g., AIM2 inflammasome upregulation [28]) and enabling temporal control of intervention [29]. In JEV-infected neurons, the activated products of the NLRP3 inflammasome and DAMPs released by pyroptotic cells collectively form a robust positive feedback loop, which in turn profoundly promotes the expression of inflammasome components. When MCC950

potently inhibits NLRP3 activation, it simultaneously disrupts this self-amplifying signaling loop. Thus, the observed reduction in expression is not a direct result of MCC950 inhibiting NF-κB, but rather an indirect consequence of its successful blockade of NLRP3 activity, which eliminates a strong NLRP3-dependent positive feedback signal. This model is supported by a growing body of recent literature, which indicates that inflammasome activity is tightly linked to the sustained expression of inflammatory genes [30]. In the experimental context, JEV provides a persistent and intense stimulus, making this positive feedback effect particularly prominent. The incomplete therapeutic efficacy suggests that JEV-induced neuroinflammation involves redundant signaling networks, including PI3K/AKT-mediated pro-inflammatory pathways [31]. In *vitro* studies further revealed that microglial activation occurred independently of NLRP3 inflammasome activation, indicating that it is not dependent on the NLRP3/caspase-1/GSDMD signaling pathway. This suggests that even if neurons are protected from viral infection, activated microglia can still drive persistent neuroinflammation [32]. Future studies should define optimal NLRP3 inhibition timing, explore combination therapies targeting both pyroptosis and viral replication, and investigate interactions with antiviral pathways (e.g., RIG-I signaling [33]). In the meantime, targeting the excessive activation of microglia may provide an additional strategy to protect the nervous system from viral injury, especially when combined with NLRP3 inflammasome inhibition.

The view that "pyroptosis is an immune defense mechanism of the body after infection" further emphasizes that pyroptosis is an adaptive manifestation of inflammatory changes caused by infection [34]. Viral infections trigger cellular immune responses, including pyroptosis, which may function as a defense mechanism against excessive viral replication [35]. Early antiviral responses can mitigate pyroptosis, but persistent inflammation—driven by unresolved viral infection—ultimately precipitates this lytic cell death pathway [36,37]. In this study, IL-1β, IL-18, and TNF-α were measured as inflammatory markers. IL-1β and IL-18, known effectors of pyroptosis [38], initially decreased post-infection, likely due to host antiviral responses diverting cellular resources. However, late-stage viral proliferation reinduced their expression, coinciding with exacerbated inflammation and massive cell death, suggesting JEV infection progresses as a protracted, persistent process. While IL-18 is processed by caspase-1/ASC, its functional activation requires additional steps: Binding to IL-18Rα/IL-18Rβ and MyD88-dependent signaling, making it a less direct marker of NLRP3 inflammasome activity. As noted in Dong et al., pro-IL-18 processing by caspase-1 is less dependent on exosite interactions, complicating its interpretation as a single-pathway readout [39]. We are more focused on IL-1β protein detection. Our results demonstrate that NLRP3 inflammasome activation drives significant IL-1β expression. However, NLRP3 inhibition failed to reduce IL-1β release, indicating a critical paradox: the CNS source of IL-1β extends beyond neurons [40]. This finding suggests glial cells—particularly microglia and astrocytes—may serve as alternative IL-1β producers during neuroinflammation.

Given the significant uncertainty in the process of differentiating primary cells into DA neurons, the use of primary cells for further targeted inflammation research is fraught with challenges and potential inaccuracies [41]. Consequently, we decided not to utilize primary cells in this regard. In addition, considering that the current method of generating dopaminergic neurons from the induced pluripotent stem cells from Parkinson's disease patients has limitations and does not meet the specific experimental design and research objectives of our study, it is rendered inapplicable for our research. Therefore, in order to overcome these obstacles and advance our research in understanding the mechanisms related to infection to JEV infection, one of the directions of our research group is to explore and develop new techniques. These techniques aim to generate specific abnormal dopaminergic neuron cell lines, which are expected to provide more accurate and targeted research materials for our in-depth study.

In this study, we focused exclusively on neurons primarily responsible for neurotransmitter secretion, without addressing the cross-reactions among neurotransmitters. While we sought to explore the potential pathological responses of microglia to dopaminergic neuron damage, our investigation was confined to the NLRP3/ASC/GSDMD signaling pathway, lacking more direct evidence to substantiate the interaction between microglia and neurons. Accordingly, in subsequent

research, we will prioritize identifying the key sites and temporal windows where microglia exert their effects during viral neuronal infection. This work aims to provide critical evidence for the development of inhibitors targeting excessive microglial activation and neuronal inflammation. Furthermore, these inhibitors are expected to offer novel insights into the development of prevention and control strategies for Japanese encephalitis virus and other mosquito-borne diseases. In turn, this will guide the rational allocation of public health resources, facilitate the formulation of personalized protection plans for high-risk populations prone to inflammatory storms, improve prevention efficacy, and reduce the associated medical burden.

## Conclusions

In this study, the damaging effect of JEV infection on dopaminergic neurons was examined *in vivo* and *in vitro*, and it was found that JEV induced pyroptosis during the clinical symptom stage and activated the NLRP3/caspase-1/GSDMD pathway while targeting dopaminergic neuron (Fig 9). These findings indicate that the motor dysfunction caused by JEV involves the targeting of dopaminergic neurons, and that inhibiting pyroptosis may alleviate this phenomenon to some extent, which may be an effective strategy for the treatment of JEV infection.

Fig 9 is adapted from "Pathogenicity and virulence of Japanese encephalitis virus: Neuroinflammation and neuronal cell damage" by Ashraf et al, used under CC BY 4.0 [4].

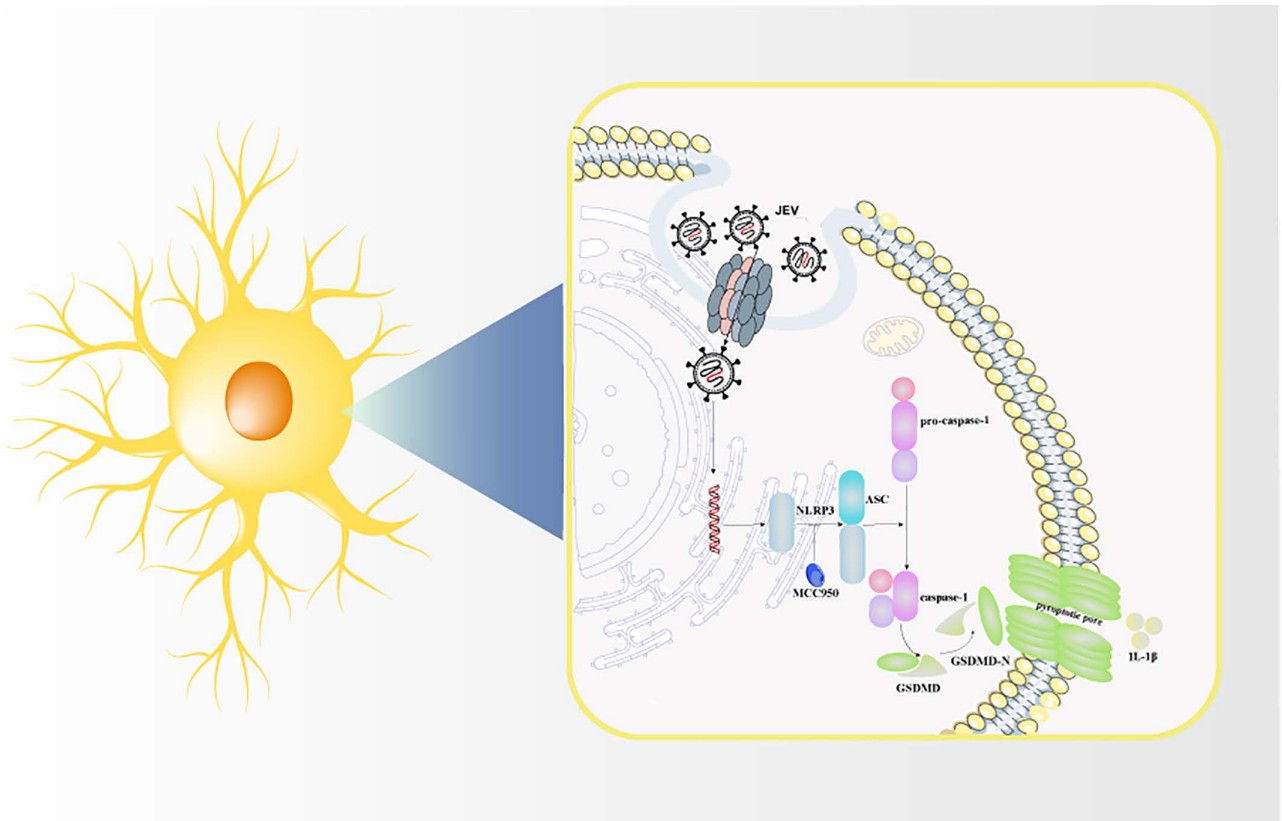

**Fig 9. Japanese encephalitis virus activates the NLRP3/caspase-1/GSDMD signaling pathway in dopaminergic neurons.**

## Supporting information

**S1 Table. Excel sheet containing raw data of Figs.** Fig 1C including positive cells of tyrosine hydroxylase$^+$ cell and JEV-E$^+$ cell at MOCK, 3dpi, 6dpi and 9dpi. Fig 1D including mean fluorescence intensity of tyrosine hydroxylase$^+$ cell and JEV-E$^+$ cell at MOCK, 3dpi, 6dpi and 9dpi. Fig 2B including positive cells of NLRP3$^+$ cell and ASC$^+$ cell at MOCK, 3dpi, 6dpi and 9dpi. Fig 2D including positive cells of caspase-1$^+$cell with tyrosine hydroxylase$^+$ cell at MOCK, 3dpi, 6dpi and 9dpi; mean fluorescence intensity of caspase-1$^+$cell with tyrosine hydroxylase$^+$ cell at MOCK, 3dpi, 6dpi and 9dpi; GSDMD$^+$ cell with tyrosine hydroxylase$^+$ cell at MOCK, 3dpi, 6dpi and 9dpi; mean fluorescence intensity of GSDMD$^+$ cell with tyrosine hydroxylase$^+$ cell at mock, 3dpi, 6dpi and 9dpi; Fig 2E including Pearson coefficients of JEV$^+$/TH$^+$, TH$^+$/caspase-1$^+$, TH$^+$/GSDMD$^+$ and JEV$^+$/IBA-1$^+$ cell at MOCK, 3dpi, 6dpi and 9dpi. Fig 3A including qPCR of mice about NLRP3, ASC, caspase-1, GSDMD, IL-1β, IL-18, TNF-α and HMGB1 at MOCK, 3dpi, 6dpi and 9dpi. Fig 3C including Western blot of mice about NLRP3, ASC, pro-caspase-1, caspase-1, GSDMD-FL, GSDMD-N, pro-IL-1β and IL-1β at MOCK, 3dpi, 6dpi and 9dpi. Fig 4B including extracellular LDH of dopaminergic neurons (MN9D cell) at MOCK, 12hpi, 24hpi, 36hpi, 48hpi and 60hpi. Fig 4C including JEV copies of dopaminergic neurons (MN9D cell) at MOCK, 12hpi, 24hpi, 36hpi, 48hpi and 60hpi. Fig 5A including qPCR of dopaminergic neurons (MN9D cell) about NLRP3, ASC, caspase-1, GSDMD, IL-1β, IL-18 and TNF-α at MOCK, 12hpi, 24hpi, 36hpi, 48hpi and 60hpi. Fig 5C including Western blot of dopaminergic neurons (MN9D cell) about NLRP3, ASC, pro-caspase-1, caspase-1, GSDMD-FL, GSDMD-N, pro-IL-1β and IL-1β at MOCK, 12hpi, 24hpi, 36hpi, 48hpi and 60hpi. Fig 7B including NESS score of mice in group DMEM, group MCC950, group JEV and group JEV + MCC950 from day1 to day 10. Fig 7E including qPCR of mice about NLRP3, ASC, caspase-1, GSDMD, IL-1β, IL-18, TNF-α and HMGB1 at MOCK, MCC950, 3dpi of JEV, 3dpi of JEV + MCC950, 6dpi of JEV, 6dpi of JEV + MCC950, 9dpi of JEV and 9dpi of JEV + MCC950. Fig 7G including Western blot of mice about NLRP3, ASC, pro-caspase-1, caspase-1, GSDMD-FL, caspase-1, pro-IL-1β and IL-1β at MOCK, MCC950, 3dpi of JEV, 3dpi of JEV + MCC950, 6dpi of JEV, 6dpi of JEV + MCC950, 9dpi of JEV and 9dpi of JEV + MCC950. Fig 8B including positive cells of IBA-1$^+$ cell and JEV-E$^+$ cell at MOCK, 3dpi, 6dpi and 9dpi; mean fluorescence intensity of IBA-1$^{++}$ cell and JEV-E$^+$ cell at MOCK, 3dpi, 6dpi and 9dpi. Fig 8D including qPCR of microglial (BV2 cell) about NLRP3, ASC, caspase-1, GSDMD, IL-1β, IL-18, TNF-α and HMGB1 at MOCK, 12hpi, 24hpi, 36hpi, 48hpi and 60hpi. Fig 8F including Western blot of microglial (BV2 cell) about NLRP3, ASC, pro-caspase-1, caspase-1, GSDMD-FL and GSDMD-N at MOCK, 12hpi, 24hpi, 36hpi and 48hpi.
(XLSX)

**S1 Fig. Renamed 2cbca.**
(TIF)

## Acknowledgments

The authors would sincerely like to thank Zelin Liu and Mingxing Gao for their contributions to the data collection for this study. We would like to thank the State Key Laboratory of Agricultural Microbiology Core Facility for assisting with structured illumination microscopy. We are also sincerely grateful to all the laboratory members in Dr. Cao laboratory for their helpful present and suggestions.

## Author contributions

**Conceptualization:** Xiaoyan Guo, Changqin Gu.

**Formal analysis:** Xiaoyan Guo.

**Funding acquisition:** Guofu Cheng, Changqin Gu.

**Investigation:** Xiaoyan Guo, Siyuan Lu, Zhiwei Huang, Jiahuan Li.

**Project administration:** Changqin Gu.

**Resources:** Guofu Cheng, Changqin Gu.

**Supervision:** Guofu Cheng, Wanpo Zhang, Xueying Hu, Changqin Gu.

**Validation:** Xiaoyan Guo, Siyuan Lu, Zhiwei Huang, Jiahuan Li.

**Visualization:** Xiaoyan Guo, Siyuan Lu, Zhiwei Huang, Jiahuan Li.

**Writing – original draft:** Xiaoyan Guo, Changqin Gu.

**Writing – review & editing:** Xiaoyan Guo, Guofu Cheng, Wanpo Zhang, Xueying Hu, Changqin Gu.

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
