## [Decision Letter · Decision Letter 0]

20 Oct 2025

Japanese encephalitis virus activates the NLRP3/caspase-1/GSDMD signaling pathway in dopaminergic neurons

Dear Dr. Gu,

Thank you for submitting your manuscript to PLOS Neglected Tropical Diseases. After careful consideration, we feel that it has merit but does not fully meet PLOS Neglected Tropical Diseases's publication criteria as it currently stands. Therefore, we invite you to submit a revised version of the manuscript that addresses the points raised during the review process.

Please submit your revised manuscript within 60 days Dec 19 2025 11:59PM. If you will need more time than this to complete your revisions, please reply to this message or contact the journal office at plosntds@plos.org. Please include the following items when submitting your revised manuscript:

We look forward to receiving your revised manuscript.

Kind regards,

Richard A. Bowen

Academic Editor

Abdallah Samy

Section Editor

Shaden Kamhawi

co-Editor-in-Chief

Paul Brindley

co-Editor-in-Chief

**Additional Editor Comments:**

Your manuscript has been reviewed and there are reviewer comments that we would like you to respond to and perhaps modify your manuscript correspondingly. Thank you.

**Journal Requirements:**

At this stage, the following Authors/Authors require contributions: Xiaoyan GUO, Siyuan Lu, Zhiwei Huang, Jiahuan Li, Guofu Cheng, Xueying Hu, Wanpo Zhang, and Changqin Gu. Please ensure that the full contributions of each author are acknowledged in the "Add/Edit/Remove Authors" section of our submission form.

2) We note that your Data Availability Statement is currently as follows: "All relevant data are within the manuscript and its Supporting Information files.". Please confirm at this time whether or not your submission contains all raw data required to replicate the results of your study. Authors must share the “minimal data set” for their submission. PLOS defines the minimal data set to consist of the data required to replicate all study findings reported in the article, as well as related metadata and methods (https://journals.plos.org/plosone/s/data-availability#loc-minimal-data-set-definition).

3) Please send a completed 'Competing Interests' statement, including any COIs declared by your co-authors. If you have no competing interests to declare, please state "The authors have declared that no competing interests exist". Otherwise please declare all competing interests beginning with the statement "I have read the journal's policy and the authors of this manuscript have the following competing interests"

**Reviewers' Comments:**

Reviewer's Responses to Questions

**Key Review Criteria Required for Acceptance?**

**Methods**

-Are the objectives of the study clearly articulated with a clear testable hypothesis stated?

-Is the study design appropriate to address the stated objectives?

-Is the population clearly described and appropriate for the hypothesis being tested?

-Is the sample size sufficient to ensure adequate power to address the hypothesis being tested?

-Were correct statistical analysis used to support conclusions?

-Are there concerns about ethical or regulatory requirements being met?

Reviewer #1: (No Response)

Reviewer #2: The objectives of the study are clearly articulated in last paragraph of the introduction section, an appropriate study design, and statistical analyses were implemented. A clear ethical statement was also provided.

**Results**

-Does the analysis presented match the analysis plan?

-Are the results clearly and completely presented?

-Are the figures (Tables, Images) of sufficient quality for clarity?

Reviewer #1: (No Response)

Reviewer #2: Overall results are presented clearly and completely.

A few minor suggestions:

Line 116: Please provide the full term for this abbreviation FRET/FCCS.

Figure 1 C and D: The color intensity between 3 dpi and 6 dpi is a bit hard to interpret. Maybe use a more distinct variation? This goes for all the similar graphs throughout the manuscript.

**Conclusions**

-Are the conclusions supported by the data presented?

-Are the limitations of analysis clearly described?

-Do the authors discuss how these data can be helpful to advance our understanding of the topic under study?

-Is public health relevance addressed?

Reviewer #1: (No Response)

Reviewer #2: The conclusions are supported by the presented data.

Would have been better if some of the study limitations were mentioned.

Also suggest including a final sentence that ties everything together and ultimately emphasizes how these findings are relevant to public health.

**Editorial and Data Presentation Modifications?**

Reviewer #1: (No Response)

Reviewer #2: Line 44: Would be great if a reference/s could be provided to back the number of reported cases worldwide.

**Summary and General Comments**

Reviewer #1: In this manuscript, the authors have investigated both in vivo and in vitro NLRP3 inflammasome activation in neurons following JEV infection. While this study has some interests, I have major criticisms:

-Figure 2C. In this figure, is it caspase-1 or activated caspase-1 that is detected in IF? Likewise, is it GSDMD or cleaved GSDMD that is detected? Indeed the authors wrote line 164 “Collectively, these findings demonstrate that the NLRP3 inflammasome mediates classical pyroptosis in JEV-infected dopaminergic neurons, linking Caspase-1 activation and GSDMD upregulation to neuroinflammatory pathogenesis.” Actually, caspase-1 (pro-caspase-1) and GSDMD upregulation is not a marker of NLRP3 inflammasome activation per se, since caspase-1 has to be matured through self-processing to process GSDMD into its pro-pyroptotic form and to cleave pro-Il1b and pro-IL-18 into their active form.

-In figure 3, qPCR and WB were performed on mock-infected and JEV-infected mouse brains where other cells (glial and microglial cells) than neurons are present so that the presented results may be not specific of neurons… While during NLRP3 inflammasome activation, it is well documented that NLRP3 and pro-IL1B expression is increased following the priming as a consequence of NF-kB activation, the transcriptional increase of ASC, caspase-1 or GSDMD is known to be much less marked than for NLRP3 and pro-IL-1β. I’m therefore surprised to observe here such an increase in ASC expression and GSDMD following infection. In contrast and surprisingly, increase in pro-IL-1b expression is rather week.

-In figure 5, like in figure 2C, which form of caspase-1 and GSDMD is detected? During NLRP3 inflammasome activation, ASC has been shown to form “specks” but here, it does not seem to be the case. Why?

-In figure 6, secretion of IL-1b and IL-18 should be assessed in JEV-infected MN9D cells by ELISA.

-Figure 6A. NLRP3 activation leads to the recruitment of the adaptor ASC and pro-caspase-1 to form the inflammatory complex. However, their basal expression (amount of protein produced) is not necessarily directly increased by NLRP3 activation. What changes is their assembly into specks (ASC specks) and the cleavage/activation of caspase-1 (switch from pro-caspase-1 to active caspase-1 p20/p10). Upon inflammasome activation, caspase-1 cleaves GSDMD into its N-terminal fragment, which forms membrane pores and triggers pyroptosis. Here too, it is not necessarily the total expression of GSDMD that increases, but its cleavage into the active form (GSDMD-N).

Hence, again, the fact that JEV infection increases expression of NLRP3, ASC, caspase-1, GSDMD, pro-IL-1b/-18 is not a formal marker for NLRP3 inflammasome activation, rather a marker for NF-kB activation (likely as consequence of TLRs or RLRs stimulation) as confirmed by the level of TNFa.

-Figure 6B. WBs are really poor quality. Cleaved GSDMD and IL-1b are detected 12hrs post-infection, not after 24h but detected again after 36hrs, this is not logical. Likewise, there are variations in the expression of NLRP3, ASC and GSDMD that are difficult to explain.

-Figure 7. MCC950 is a specific inhibitor of NLRP3 inflammasome by preventing conformational activation of NLRP3. It Inhibits the formation of the NLRP3-ASC-caspase-1 complex reducing caspase-1 cleavage, thus the maturation and secretion of IL-1β and IL-18, as well as GSDMD-dependent pyroptosis. Hence, I do not understand why MCC950 inhibits expression of NLRP3, ASC, procaspase-1, GSDMD as observed by qPCR and WB. In your hands, MCC950 seems to act like a NF-kB inhibitor… this is so puzzling.

-WBs in Figure 8E are not convincing.

Minor comments:

Do dopaminergic neurons produce IL-1b or IL-18?

Line 37, I guess that the authors meant « Pyroptosis » rather than “necroptosis”

Reviewer #2: Overall, this is a well-written paper presenting exciting and impactful findings.

Wishing the researchers the very best moving forward.

PLOS authors have the option to publish the peer review history of their article (what does this mean? ). If published, this will include your full peer review and any attached files.

**Do you want your identity to be public for this peer review?** For information about this choice, including consent withdrawal, please see our Privacy Policy .

Reviewer #1: **Yes:** Damien Arnoult

Reviewer #2: No

**Figure resubmission:**
---

## [Editor Report · Decision Letter 1]

11 Jan 2026

Dear Dr Gu,

We are pleased to inform you that your manuscript 'Japanese encephalitis virus activates the NLRP3/caspase-1/GSDMD signaling pathway in dopaminergic neurons' has been provisionally accepted for publication in PLOS Neglected Tropical Diseases.

Best regards,

Richard A. Bowen, DVM PhD

Academic Editor

Abdallah Samy

Section Editor

Shaden Kamhawi

co-Editor-in-Chief

Paul Brindley

co-Editor-in-Chief

Thank you for the thoughtful and appropriate responses to reviewer comments. I think the modifications you made significantly clarify and improve your manuscript.

---

## [Editor Report · Acceptance letter]

Dear Dr Gu,

We are delighted to inform you that your manuscript, "Japanese encephalitis virus activates the NLRP3/caspase-1/GSDMD signaling pathway in dopaminergic neurons," has been formally accepted for publication in PLOS Neglected Tropical Diseases.

Best regards,

Shaden Kamhawi

co-Editor-in-Chief

Paul Brindley

co-Editor-in-Chief
